# Neuropeptide Y-expressing dorsal horn inhibitory interneurons gate spinal pain and itch signalling

Kieran A Boyle[1†], Erika Polgar[1†], Maria Gutierrez-Mecinas[1†], Allen C Dickie[1†], Andrew H Cooper[1], Andrew M Bell[1], Evelline Jumolea[1], Adrian Casas-Benito[1], Masahiko Watanabe[2], David I Hughes[1], Gregory A Weir[1], John S Riddell[1], Andrew J Todd[1]*

[1]School of Psychology and Neuroscience, College of Medical, Veterinary and Life Sciences, University of Glasgow, Glasgow, United Kingdom; [2]Department of Anatomy, Hokkaido University School of Medicine, Sapporo, Japan

*For correspondence:
Andrew.Todd@glasgow.ac.uk

†These authors contributed equally to this work

Competing interest: The authors declare that no competing interests exist.

**Abstract** Somatosensory information is processed by a complex network of interneurons in the spinal dorsal horn. It has been reported that inhibitory interneurons that express neuropeptide Y (NPY), either permanently or during development, suppress mechanical itch, with no effect on pain. Here, we investigate the role of interneurons that continue to express NPY (NPY-INs) in the adult mouse spinal cord. We find that chemogenetic activation of NPY-INs reduces behaviours associated with acute pain and pruritogen-evoked itch, whereas silencing them causes exaggerated itch responses that depend on cells expressing the gastrin-releasing peptide receptor. As predicted by our previous studies, silencing of another population of inhibitory interneurons (those expressing dynorphin) also increases itch, but to a lesser extent. Importantly, NPY-IN activation also reduces behavioural signs of inflammatory and neuropathic pain. These results demonstrate that NPY-INs gate pain and itch transmission at the spinal level, and therefore represent a potential treatment target for pathological pain and itch.

## eLife assessment

Boyle et al identify Npy-expressing dorsal horn neurons as powerfully inhibiting pain and itch under normal and pathological conditions. The **valuable** data are **convincing**, and the effect sizes are robust and directly challenge previous work.

## Introduction

The spinal dorsal horn represents the entry point into the CNS for somatosensory information from the trunk and limbs. This information is relayed via projection cells to supraspinal sites, where it leads to perceptions, including pain and itch (*Abraira and Ginty, 2013*; *Todd, 2010*; *Mishra and Hoon, 2015*). However, projection cells represent only ~1% of dorsal horn neurons, with the vast majority comprising excitatory and inhibitory interneurons that are arranged into local microcircuits that process somatosensory information. Altered function of these circuits contributes to chronic pain and pruritus (itch) (*Braz et al., 2014*; *Cevikbas and Lerner, 2020*). Dysregulation of inhibitory circuits has attracted particular interest, as broad disruption of spinal inhibitory signalling produces behaviours reminiscent of symptoms seen in patients suffering from chronic pain or pruritus (*Beyer et al., 1985*; *Foster et al., 2015*; *Sivilotti and Woolf, 1994*; *Yaksh, 1989*).

We have described a molecular classification scheme that assigns the inhibitory interneurons in mouse superficial dorsal horn (SDH; laminae I and II) to five largely non-overlapping populations, based on expression of calretinin (CR), parvalbumin (PV), neuronal nitric oxide synthase (nNOS), dynorphin and galanin (Dyn/Gal), or neuropeptide Y (NPY) (*Boyle et al., 2017*). This scheme has since been validated and extended by large-scale transcriptomic studies (*Häring et al., 2018*; *Sathyamurthy et al., 2018*). These molecularly defined interneuron populations appear to be functionally distinct as they display differential activation profiles in response to noxious and innocuous stimuli (*Häring et al., 2018*; *Sathyamurthy et al., 2018*; *Polgár et al., 2013*). A major advantage of this approach is that it allows investigation of the function of different populations through targeted manipulation with techniques such as chemogenetics, optogenetics, and toxin-mediated silencing or ablation. Studies of this type have implicated the PV interneurons in preventing tactile allodynia (*Boyle et al., 2019*; *Petitjean et al., 2015*), the nNOS interneurons in gating both mechanical and thermal inputs (*Huang et al., 2018*), and the Dyn/Gal population in suppressing mechanical pain and pruritogen-evoked itch (*Huang et al., 2018*; *Duan et al., 2014*). Ablation or silencing of dorsal horn NPY-lineage neurons (i.e. cells that express NPY transiently during development or persistently into adulthood) has been reported to cause spontaneous itching behaviours and enhancement of touch-evoked (mechanical) itch, without affecting pruritogen-evoked itch or pain behaviours (*Bourane et al., 2015*). This has led to the view that the main function of the NPY cells is suppression of mechanical itch (*Koch et al., 2018*; *Pan et al., 2019*; *Acton et al., 2019*; *Liu et al., 2019*; *Chen et al., 2020*; *Chen and Sun, 2020*). This limited role for NPY interneurons is surprising for several reasons: (1) they account for one-third of all inhibitory interneurons in SDH *Boyle et al., 2017*; (2) they innervate a population of nociceptive projection cells that belong to the anterolateral system (ALS) (*Polgár et al., 2013*; *Cameron et al., 2015*; *Iwagaki et al., 2016*; *Kókai et al., 2022*); (3) while mechanical itch is restricted to hairy skin, NPY-expressing neurons are present throughout the dorsal horn, including areas innervated from glabrous skin (*Boyle et al., 2017*; *Polgár et al., 2011*); and (4) NPY itself has a role in modulating chronic pain (*Solway et al., 2011*). As noted above, the approach used by *Bourane et al., 2015* targeted a broad population of inhibitory interneurons that express NPY during development, as well as those that express NPY in adulthood. *Tashima et al., 2021* recently attempted to target NPY-INs by injecting adeno-associated viruses (AAVs) with a *Npy* promoter into the rat spinal cord. However, expression was largely restricted to lamina IIo (even though many NPY cells are found in other laminae) and fewer than half of the targeted cells contained either NPY or its mRNA. Therefore, the role of those dorsal horn interneurons that continue to express NPY (NPY-INs) remains unclear.

Here, we use intraspinal injections of AAVs carrying Cre-dependent expression cassettes into young adult Npy-Cre mice to target dorsal horn NPY-INs. We demonstrate that this technique can be used to manipulate inhibitory interneurons that express NPY in adulthood, while avoiding those cells that transiently express NPY during development. We show that chemogenetic activation of dorsal horn NPY-INs suppresses acute mechanical and thermal nocifensive behaviours, as well as those resulting from pruritogen-evoked itch, and reduces activity in spinal networks that process nociceptive and pruritoceptive information. Furthermore, NPY-IN activation abolishes mechanical and thermal hypersensitivity in models of inflammatory and neuropathic pain. Finally, we show that silencing of NPY-INs results in spontaneous itch and an exaggerated response to pruritogens, and that this depends on a circuit involving GABAergic input from NPY-INs to excitatory interneurons that express the gastrin-releasing peptide receptor (GRPR). Together these results demonstrate that dorsal horn NPY-INs have a far broader role than previously suggested, since they gate transmission of nociceptive and pruriceptive information. They therefore represent a potential target for the development of new treatments for pain and itch.

## Results

### Cre-dependent AAV injections in young adult Npy-Cre mice target dorsal horn inhibitory NPY interneurons and avoid transient NPY-expressing cells

We initially assessed the suitability of using intra-spinal injection of AAVs encoding Cre-dependent constructs in RH26 Npy-Cre mice (the line used by *Bourane et al., 2015*) to target NPY-INs in the dorsal horn. We first performed RNAscope fluorescent in situ hybridisation (FISH) to compare *Cre* and

*Npy* mRNA expression in lumbar spinal cord sections from young adult Npy-Cre mice. Across laminae I–III 91.6% ± 0.3% of cells classed as *Cre*-positive cells were also *Npy*-positive, and these accounted for 62.1% ± 0.6% of *Npy*-positive cells, demonstrating that Cre expression in the Npy-Cre line faithfully captures the majority of NPY-INs in the adult dorsal horn (***Figure 1A, B***). Accordingly, injection of either AAV.flex.tdTomato (tdTom) or AAV.flex.eGFP (both serotype 1 with CAG promoter, see Key Resources Table) into the lumbar dorsal horn of adult Npy-Cre mice (***Figure 1C***) resulted in fluorescent protein (FP) expression matching that previously reported for NPY neurons (***Boyle et al., 2017***; ***Iwagaki et al., 2016***), with cell bodies concentrated in laminae I–III (***Figure 1D***). The great majority of FP-expressing neurons in laminae I–III were immunoreactive (IR) for NPY (78.5% ± 3.6%), and these accounted for 74.6% ± 1.9% of the NPY-IR neurons in this area (***Figure 1D, E***).

We then crossed Npy-Cre mice with the Cre-dependent reporter line Ai9 (to label all NPY-lineage neurons with tdTomato) and injected AAV.flex.eGFP into the lumbar dorsal horn of these mice, to target cells that continued to express NPY (***Figure 1F*** and ***Figure 1—figure supplement 1A***). In these animals, tdTom-positive cells were seen throughout the dorsal horn, and were much more numerous than eGFP-expressing cells in the region of the injection site (***Figure 1G***). All tdTom-labelled cells were IR for the transcription factor Pax2 (***Figure 1—figure supplement 1B***), which is expressed by all dorsal horn inhibitory neurons in rodents (***Foster et al., 2015***; ***Larsson, 2017***), and these accounted for 40.8% ± 10.0% of Pax2 cells in laminae I–III. Virtually, all eGFP-expressing neurons within this region were tdTom-positive (97.8% ± 0.2%), but these only accounted for 51.1% ± 3.5% of the tdTom-positive population (***Figure 1H, I***). In agreement with the data presented above, the great majority tdTom+;eGFP+ neurons were found to be NPY-IR (85.2% ± 0.2%), and these accounted for 67.0% ± 6.3% of the NPY-IR cells in laminae I–III. In contrast, only 32.1% ± 2.1% of the tdTom+;eGFP-negative cells displayed NPY immunoreactivity, corresponding to just 24.1% ± 2.7% of all NPY-IR interneurons (***Figure 1H, I***). Overall, 58.3% ± 0.9% of tdTom+ neurons were NPY-IR.

These results suggest that transient Cre expression occurs in a broad population of dorsal horn inhibitory interneurons in Npy-Cre mice, presumably driven by NPY expression during development. However, Cre expression in adult mice occurs in a much more restricted population of inhibitory interneurons that express NPY persistently. To characterise these cells in relation to the neurochemical populations of inhibitory interneurons that we had identified in the SDH (***Boyle et al., 2017***), we focussed our analysis on laminae I–II and compared the expression of the different neurochemical markers between the tdTom+;eGFP+ and tdTom+;eGFP-negative cells. Within this region, 85.8% ± 2.5% of tdTom+;eGFP+ cells were NPY-IR, while approximately 10%, 3%, and 1% expressed galanin, nNOS, or PV, respectively (***Figure 1—figure supplement 1C***). This is in good agreement with the degree of overlap between these markers and NPY that we have previously described (***Boyle et al., 2017***). Surprisingly, 23.7% ± 7.0% of tdTom+;eGFP+ cells expressed CR (***Figure 1—figure supplement 1C***), which has previously been reported to show minimal overlap with NPY-IR (***Smith et al., 2015***). Nonetheless tdTom+;eGFP+ cells are largely restricted to the NPY+ population of inhibitory interneurons. In contrast, tdTom+;eGFP-negative neurons were much more broadly spread across four of the populations, with approximately 28% expressing NPY, 44% expressing CR, and 24% each expressing galanin or nNOS, although virtually none (1.4% ± 1.4%) expressed PV (***Figure 1—figure supplement 1C***).

Because we intended to use Cre-dependent expression of the excitatory DREADD hM3Dq (***Foster et al., 2015***; ***Huang et al., 2018***) to activate NPY-INs, we also assessed targeting of this receptor to the appropriate interneurons following injection of AAV.flex.hM3Dq-mCherry into the lumbar dorsal horn of adult Npy-Cre mice (***Figure 1—figure supplement 1D***). hM3Dq-mCherry-expressing cells in the SDH displayed a near-identical pattern in terms of co-localisation with inhibitory interneuron markers to that of the eGFP+ cells following AAV.flex.eGFP injection (***Figure 1—figure supplement 1E, F***). For the mCherry+ cells, 90.5% ± 6.2% co-expressed NPY (accounting for 66.1% ± 4.8% of all NPY interneurons) and there was little or no overlap with the galanin, nNOS, and PV populations (***Figure 1—figure supplement 1E, F***). Again, significant overlap was observed between hM3Dq-mCherry- and CR-positive cells, with 28.0% ± 1.5% of mCherry-labelled cells displaying CR-IR. NPY-IR was detected in 93.1% ± 3.7% of mCherry+;CR+ cells in sections co-stained with NPY antibody (***Figure 1—figure supplement 1E***), demonstrating that this represents Npy-Cre-driven recombination in interneurons co-expressing NPY and CR, rather than ectopic recombination in CR+;NPY-negative interneurons. The level of NPY-IR in the CR cells was generally very low (***Figure 1—figure supplement***

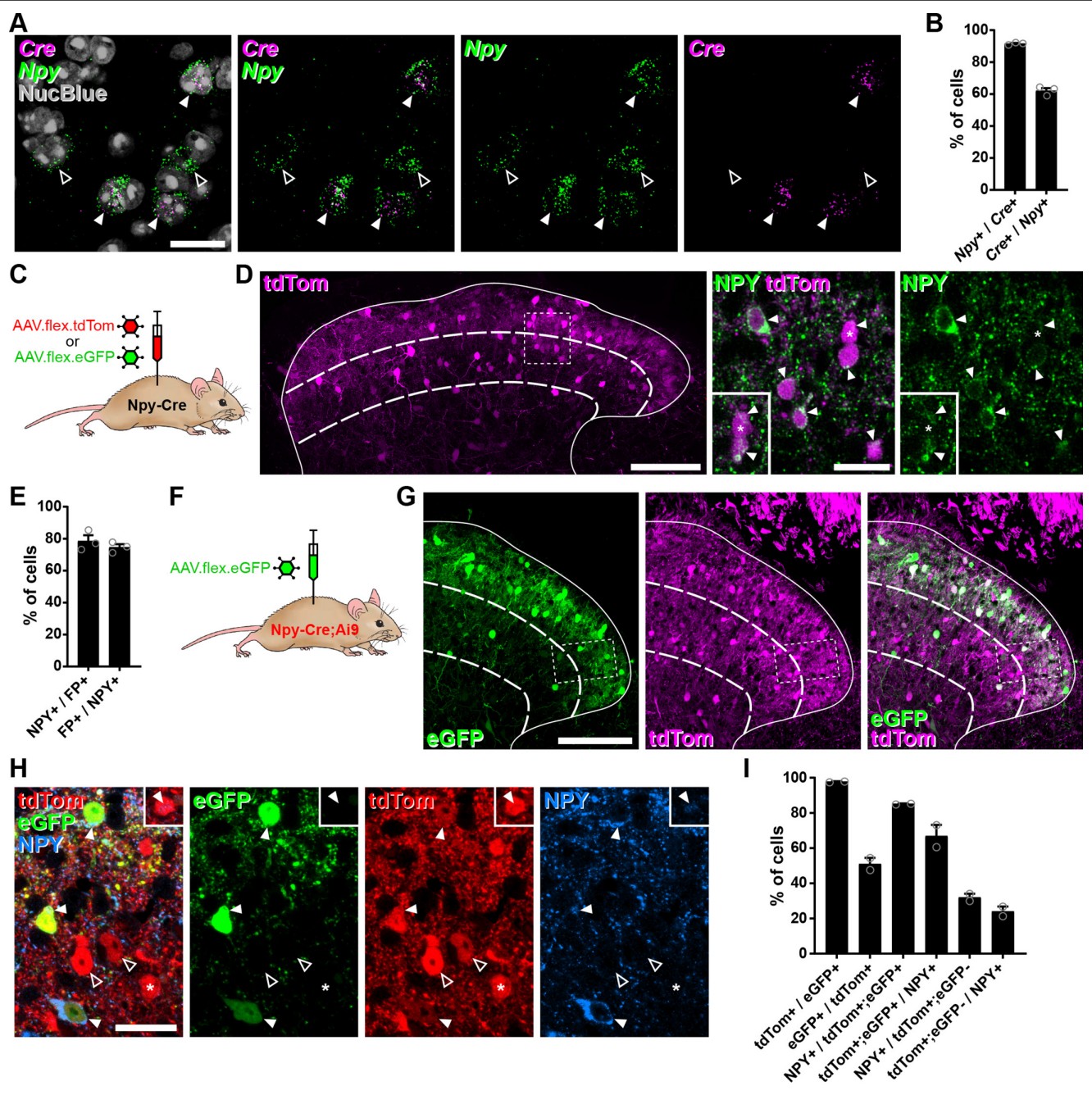

**Figure 1.** Cre-dependent adeno-associated virus (AAV) injections in young adult Npy-Cre mice target dorsal horn inhibitory neuropeptide Y (NPY) interneurons and avoid transient NPY-expressing cells. (**A**) In situ hybridisation for *Cre* (magenta) and *Npy* (green) mRNA in the mid-lumbar dorsal horn. Sections were counterstained with NucBlue (grey) to reveal nuclei. Three *Cre*-positive cells are also positive for *Npy* (filled arrowheads), and there are two cells that are positive for *Npy* only (open arrowheads). Scale bar = 20 µm. (**B**) Quantification of co-expression of *Cre* and *Npy* mRNA in laminae I–III (n = 3 mice). (**C**) The experimental approach used to generate the data presented in (**D, E**). (**D**) Co-expression of tdTomato (tdTom; magenta) and NPY (green) immunoreactivity in mid-lumbar dorsal horn of an Npy-Cre mouse injected with AAV.flex.tdTom in adulthood. Low power image shows tdTomato-positive cells throughout laminae I–III (scale bar = 100 µm). High power images (corresponding to the box in the low power image) demonstrate the high degree of co-localisation of tdTomato expression and NPY immunoreactivity (filled arrowheads; scale bar = 20 µm). Insets show clearer NPY labelling in a different z-plane for the cell marked with an asterisk and the cell immediately below it. (**E**) Quantification of co-expression of fluorescent protein (FP) and NPY in laminae I–III of Npy-Cre mice injected with AAV.flex.tdTomato or AAV.flex.eGFP at adulthood (n = 3 mice; 2 injected with AAV.flex.tdTomato and 1 with AAV.flex.eGFP). (**F**) The experimental approach taken for the data presented in (**G–I**). (**G**) Expression of tdTomato and eGFP in mid-lumbar dorsal horn of an Npy-Cre;Ai9 mouse injected with AAV.flex.eGFP in adulthood. eGFP-positive cells (green) are a subset of a broader population of tdTomato-positive cells (magenta; scale bar = 100 µm). (**H**) High power images corresponding to boxed area in (**G**) showing three

*Figure 1 continued on next page*

*Figure 1 continued*

tdTom+/eGFP+ (red and green, respectively) double-labelled cells (filled arrowheads) that are also NPY immunoreactive (blue). Open arrowheads mark two cells positive for tdTomato only. The asterisk marks a tdTomato-positive/eGFP-negative cell that is also NPY immunoreactive, though this is only apparent in a different z-plane (insets). Scale bar = 20 µm. (I) Quantification of co-expression of tdTomato, eGFP, and NPY in laminae I–III of Npy-Cre;Ai9 mice injected with AAV.flex.eGFP at adulthood (*n* = 2 mice). Solid lines in low power images in (**D, G**) denote the grey/white matter border, curved dashed lines denote the boundaries of lamina III and dashed boxes denote the regions shown in corresponding high power images. Data are shown as individual values with mean ± standard error of the mean (SEM).

The online version of this article includes the following figure supplement(s) for figure 1:

**Figure supplement 1.** Cre-dependent adeno-associated virus (AAV) injections in young adult Npy-Cre mice target dorsal horn inhibitory neuropeptide Y (NPY) interneurons and avoid transient NPY-expressing cells.

*1E*), which probably explains why this overlap was not detected previously (*Smith et al., 2015*). As expected, virtually all mCherry-labelled cells were inhibitory (97.4% ± 2.6% Pax2-positive), and these accounted for a quarter of all inhibitory interneurons in the SDH (*Figure 1—figure supplement 1F*). Crucially, no mCherry-labelled cells were observed in the ipsi- or contralateral L3, L4, or L5 DRG of four AAV.flex.hM3Dq-mCherry-injected Npy-Cre mice (data not shown), as would be expected from the lack of NPY expression in uninjured mouse DRG neurons (*Honore et al., 2000*).

Collectively, these results demonstrate that injection of AAVs encoding Cre-dependent constructs into the dorsal horn of adult Npy-Cre mice allows specific targeting of most inhibitory interneurons that persistently express NPY, and avoids capturing a broader population of inhibitory interneurons that express NPY transiently during development.

## Activation of inhibitory NPY interneurons reduces activity in dorsal horn circuits recruited by nociceptive and pruritic stimuli

We initially assessed the efficacy of our chemogenetic strategy to activate NPY-INs in Npy-Cre mice injected with AAV.flex.hM3Dq-mCherry by comparing expression of the activity marker Fos 2 hr after administration of the hM3Dq ligand clozapine-*N*-oxide (CNO) or vehicle (*Figure 2—figure supplement 1A*). Only 8.8% ± 4.7% of mCherry-labelled cells in laminae I–III displayed Fos-IR in vehicle-treated animals, but this rose dramatically to 82.9% ± 2.5% in CNO-treated mice (*Figure 2—figure supplement 1B, C*). Of these mCherry+;Fos+ cells, 87.5% ± 3.1% displayed detectable NPY immunoreactivity, and these accounted for 53.9% ± 1.2% of all NPY-IR neurons (*Figure 2—figure supplement 1D, E*). Surprisingly, we also observed a small but significant increase in the proportion of mCherry-negative cells that expressed Fos following CNO (from 2.7% ± 0.3% in vehicle-treated to 5.8% ± 1.0% in CNO-treated mice), and this increase was entirely restricted to inhibitory interneurons (*Figure 2—figure supplement 1C*). A significant proportion of these mCherry-negative;Fos+ cells were also NPY-IR (31.3% ± 4.8%; *Figure 2—figure supplement 1E*), and these may represent NPY-INs that express hM3Dq-mCherry at a level sufficient for direct CNO-mediated activation, but that is too low for immunohistochemical detection. Alternatively, they may have been indirectly recruited via disinhibition following CNO-mediated activation of hM3Dq-mCherry-expressing NPY-INs. Overall, CNO treatment resulted in Fos expression in 65.9% ± 2.8% of all NPY-INs, and these comprised 65.5% ± 3.8% of Fos-expressing cells (*Figure 2—figure supplement 1E*). In summary, intraspinal injection of AAV.flex.hM3Dq-mCherry into adult Npy-Cre mice allows chemogenetic activation of two thirds of dorsal horn NPY-INs, and a small proportion of other dorsal inhibitory interneurons.

We then assessed the ability of dorsal horn NPY-INs to suppress the transmission of pain- and itch-related information at the circuit level. Npy-Cre mice that had had intraspinal injections of AAV. flex.hM3Dq-mCherry were injected with vehicle or CNO, and then received a noxious heat (hindpaw immersion in 52°C water) or pruritic (intradermal injection of chloroquine, CQ, in the calf) stimulus ipsilateral to the viral injection under brief general anaesthesia (*Figure 2A, B*). Mice that received the pruritic stimulus were fitted with an Elizabethan collar to prevent Fos induction due to itch-related biting of the leg. Following a 2-hr survival period, mice were perfused with fixative and spinal cord sections were processed for Fos-IR. In vehicle treated animals, the noxious heat and pruritic stimuli resulted in ipsilateral Fos expression in the somatotopically relevant areas of the dorsal horn. Fos+ cells were particularly clustered in the medial half of the SDH following noxious heat, and the middle third of the SDH after CQ, as previously reported (*Bell et al., 2016*; *Gutierrez-Mecinas et al., 2017*; *Figure 2C, D*). Accordingly, analysis of Fos expression was performed within these regions of the SDH.

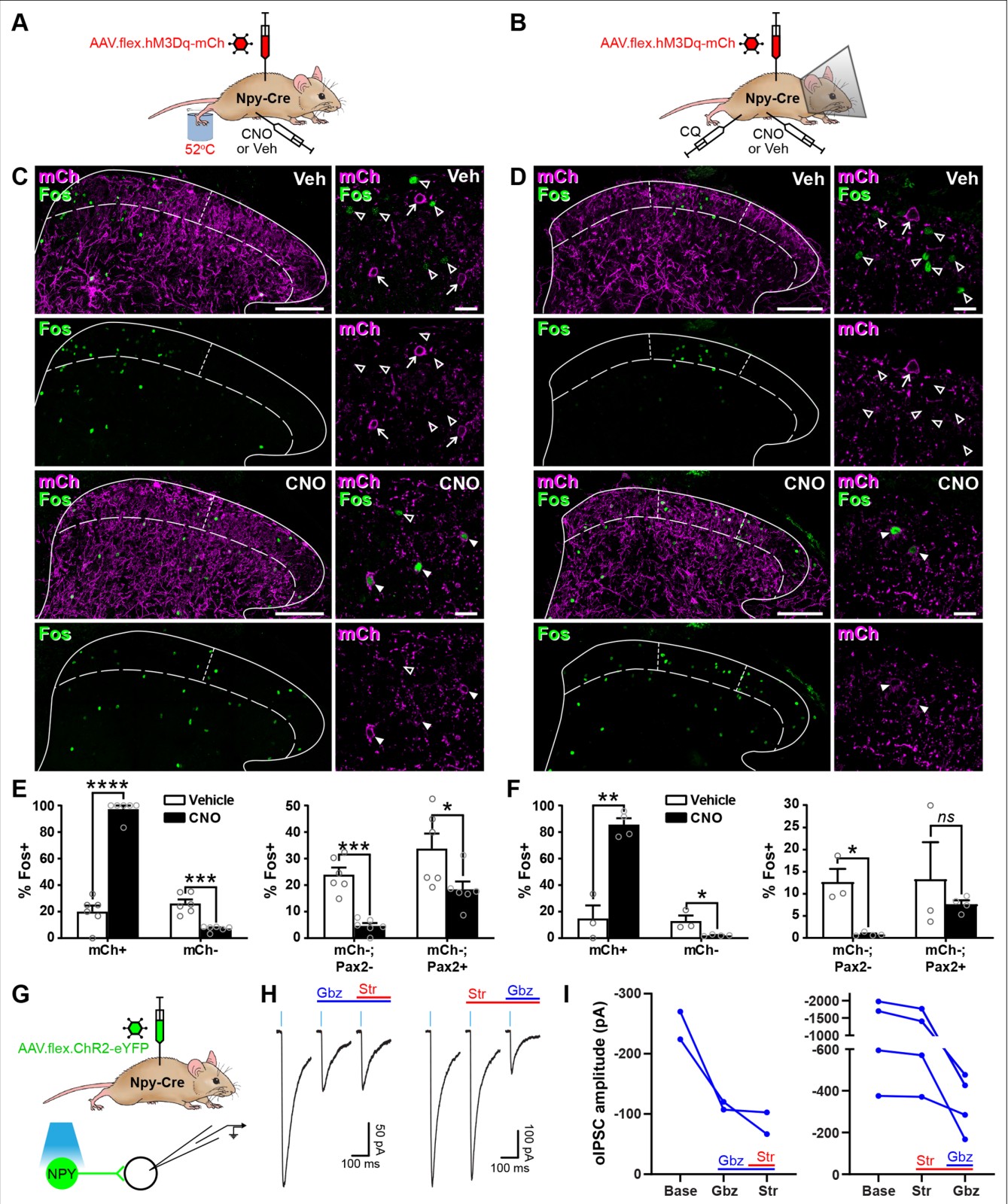

**Figure 2.** Activation of inhibitory neuropeptide Y (NPY) interneurons reduces activity in dorsal horn circuits recruited by noxious and pruritic stimuli. (**A, B**) The experimental approaches used to generate the data presented in (**C, E**) and (**D, F**), respectively. (**C, D**) Low power images show mCherry expression (mCh; magenta) and Fos immunoreactivity (green) in L4 (**C**) or L3 (**D**) dorsal horn of Npy-Cre mice that had been injected with AAV.flex. hM3Dq-mCherry and treated with vehicle control (Veh) or clozapine-*N*-oxide (CNO) 30 min prior to a noxious heat stimulus (immersion of the hindpaw

*Figure 2 continued on next page*

*Figure 2 continued*

in 52°C water; **C**) or a pruritic stimulus (intradermal injection of 100 μg chloroquine, CQ, dissolved in 10 μl phosphate-buffered saline (PBS) into the calf; **D**) and perfusion-fixed 2-hr post-stimulation. In vehicle-treated animals, Fos expression is observed in the somatotopically relevant area of the superficial laminae (left of short dashed line in **C**; between short dashed lines in **D**). In CNO-treated animals, Fos expression is observed in hM3Dq-mCherry-expressing cells, but is reduced in surrounding hM3Dq-mCherry-negative cells within the somatotopically relevant area. This is demonstrated in the high magnification images (to the right of the main image in each case), where filled arrowheads mark examples of hM3Dq-mCherry-expressing cells immunoreactive for Fos, open arrowheads mark Fos-positive cells that lack hM3Dq-mCherry and arrows mark hM3Dq-mCherry-expressing cells that are negative for Fos. Scale bars = 100 μm (low power images), 20 μm (high power images). Solid lines in (**C** and **D**) denote the grey/white matter border, curved dashed lines denote the lamina II/III border. (**E, F**) Left-hand graphs show quantification of the proportion of hM3Dq-mCherry-positive (mCh+) and -negative (mCh−) cells that display Fos immunoreactivity in vehicle- or CNO-treated mice that received a noxious heat (**E**; *n* = 3 mice for vehicle, 4 for CNO) or pruritic (**F**; *n* = 6 for both groups) stimulus. Right-hand graphs in (**E, F**) show quantification of the proportion of hM3Dq-mCherry-negative cells that display Fos immunoreactivity, separated into excitatory (Pax2−) and inhibitory (Pax2+) populations. Analyses were performed within the somatotopically relevant areas of laminae I and II. (**G**) The experimental approach used to generate the data presented in (**H, I**). (**H**) Representative optogenetically evoked IPSCs (oIPSCs) recorded in unlabelled (ChR2-eYFP-negative) cells in spinal cord slices from Npy-Cre mice that had received intraspinal injections of AAV.flex.ChR2-YFP. Recordings were made in the absence and presence of gabazine (Gbz) and strychnine (Str). Traces show an average of six stimuli, light blue bars denote period of optogenetic stimulation. Note that in the presence of high intracellular Cl⁻ concentration, IPSCs appear as inward currents. (**I**) Quantification of the mean peak amplitude of oIPSCs recorded in the absence (Base) and presence of gabazine and strychnine. Two cells were initially tested with gabazine and then strychnine, while four cells were initially tested with strychnine and then gabazine. Data are shown as mean ± standard error of the mean (SEM). *p < 0.05, **p < 0.01, ***p < 0.001, ****p < 0.0001; unpaired *t*-test with Holm–Šidák correction for multiple comparisons. Data are shown as individual values with mean ± SEM in (**E, F**) and individual values in (**I**).

The online version of this article includes the following figure supplement(s) for figure 2:

**Figure supplement 1.** Clozapine-*N*-oxide (CNO)-mediated activation of dorsal horn inhibitory neuropeptide Y (NPY) interneurons in Npy-Cre mice injected with AAV.flex.hM3Dq-mCherry.

**Figure supplement 2.** Characterisation of optogenetic activation of NPY-INs.

In CNO-treated mice there was a clear increase in the proportion of mCherry-labelled cells expressing Fos (*Figure 2C–F*), presumably due to direct chemogenetic activation of these cells (as described above). However, for both the noxious heat and pruritic stimuli, we observed a significant decrease in the proportion of mCherry-negative cells that were Fos-positive, when compared to vehicle-treated mice (noxious heat: vehicle = 26.1% ± 3.1% vs. CNO = 7.2% ± 0.8%; CQ injection: vehicle = 12.9% ± 4.2% vs. CNO = 2.1% ± 0.3%) (*Figure 2C–F*). In both cases, the decrease was largely restricted to Pax2-negative (excitatory) neurons, although a significant decrease was also observed in Pax2-positive cells in heat-stimulated mice, with a similar trend in CQ-stimulated mice (*Figure 2E, F*).

These results demonstrate that activation of NPY-INs inhibits neurons that are normally recruited by noxious or pruritic stimuli in dorsal horn circuits. To investigate the nature of this inhibition in more detail, we performed optogenetic experiments in spinal cord slices from Npy-Cre mice that had received intraspinal injections of AAV.flex.ChR2-eYFP, resulting in expression of eYFP-tagged channelrhodopsin in NPY-INs (*Figure 2G*). Short pulses of blue light reliably evoked inward currents and action potential firing in all (10/10) eYFP-ChR2+ cells tested (*Figure 2—figure supplement 2A–C*). Recordings were made from 41 ChR2-eYFP-negative cells in the SDH (*Figure 2G* and *Figure 2—figure supplement 2D*), with an optogenetically evoked postsynaptic current (oPSC) being seen in 29 cells (70.7%). In seven of the cells with oPSCs, bath application of the AMPAr and NMDAr antagonists, NBQX and D-APV, respectively, did not alter the peak amplitude of the current (baseline = −779.8 ± 267.7 pA vs. NBQX/D-APV = −756.6 ± 285.8 pA, p = 0.578, Wilcoxon matched-pairs signed rank test) (*Figure 2—figure supplement 2E, F*), demonstrating that these were not mediated by glutamate and were therefore optogenetically evoked IPSCs (oIPSCs). The GABAergic/glycinergic nature of these oIPSCs was investigated by bath application of gabazine and strychnine, respectively (in the presence of NBQX and D-APV). All oIPSCs tested (6/6) were reduced by gabazine, but not strychnine (*Figure 2H, I*), indicating that inhibition is predominantly mediated by GABA. This is consistent with the finding that NPY neurons in laminae I–III are all GABA-IR, but are not enriched with glycine (*Rowan et al., 1993*). Taken together, these findings demonstrate that NPY-INs provide a powerful GABAergic inhibitory input to surrounding dorsal horn neurons and can reduce the activation of excitatory neurons that are normally recruited by noxious or pruritic stimuli, suggesting that when activated they supress the transmission of pain- and itch-related information in the dorsal horn.

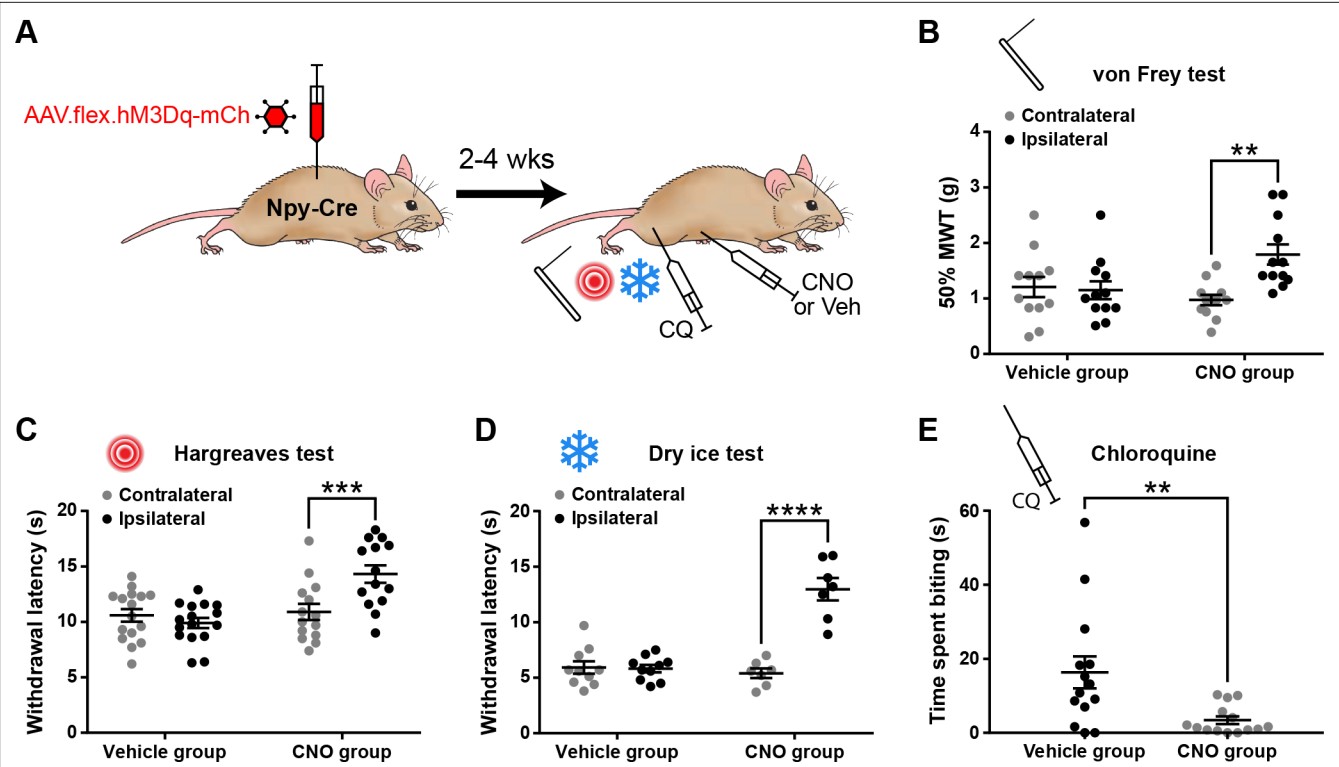

**Figure 3.** Activation of inhibitory neuropeptide Y (NPY) interneurons increases acute nociceptive thresholds and reduces pruritogen-evoked itch behaviour. (**A**) The experimental approach used to generate the data presented in (**B–E**). (**B–D**) AAV.flex.hM3Dq-mCherry spinal-injected Npy-Cre mice display an increased mechanical withdrawal threshold (MWT) (**B**; *n* = 12 vehicle group, 12 clozapine-*N*-oxide [CNO] group) and increased withdrawal latencies to radiant heat (**C**; *n* = 16 vehicle group, 14 CNO group) and to cold (**D**; *n* = 10 vehicle group, 7 CNO group) of the ipsilateral hindpaw following CNO injection, but not vehicle control injection. (**E**) AAV.flex.hM3Dq-mCherry spinal-injected mice spend significantly less time biting the calf region in the 30 min following intradermal injection of chloroquine when injected with CNO, compared to vehicle-treated controls (*n* = 14 vehicle group, 14 CNO group). Data are shown as individual values with mean ± standard error of the mean (SEM). **p < 0.01, ***p <0.001, ****p < 0.0001; repeated-measures two-way ANOVA with Šidák's post-test in (**B–D**); unpaired *t*-test in (**E**).

The online version of this article includes the following figure supplement(s) for figure 3:

**Figure supplement 1.** Injection of 5 mg/kg clozapine-*N*-oxide (CNO) does not result in off-target behavioural effects.

## Activation of inhibitory NPY interneurons increases acute nocifensive reflex thresholds and reduces pruritogen-evoked itch behaviour

We then looked for behavioural correlates of the NPY-IN-mediated suppression of dorsal horn pain and itch circuits. To do this, we assessed mechanical and thermal nocifensive reflexes, as well as CQ-induced itch behaviour, in Npy-Cre mice that had received unilateral spinal injections of AAV.flex. hM3Dq-mCherry and were then treated with vehicle or CNO (*Figure 3A*). In vehicle-treated mice, as expected, there were no significant differences between the hindpaws contralateral and ipsilateral to the AAV injection for the 50% mechanical withdrawal threshold (MWT) or for withdrawal latencies to noxious heat or cold. However, in CNO-treated mice nocifensive thresholds/latencies in the ipsilateral paw were significantly increased across all three modalities, demonstrating a generalised anti-nociceptive effect of NPY-IN activation (*Figure 3B–D*). CNO-treated mice also spent significantly less time biting the calf area in the 30 min following CQ injection than vehicle-treated controls, demonstrating a reduction in pruritogen-evoked itch upon NPY-IN activation (*Figure 3E*). It has been proposed that when used at high doses, systemic CNO may have off-target effects as a result of conversion to clozapine (*Gomez et al., 2017*). We therefore tested the effect of 5 mg/kg CNO (the dose used throughout our study) on naive wild-type mice, and found no change in mechanical or thermal nocifensive thresholds, or on locomotor performance (*Figure 3—figure supplement 1*). This confirms that the effects observed in AAV.flex.hM3Dq-mCherry-injected Npy-Cre mice are due to DREADD activation, and not the result of off-target effects of CNO. These findings show that

chemogenetic activation of dorsal horn NPY-INs has a broad anti-nociceptive effect across a range of modalities and suppresses pruritogen-evoked itch.

## Activation of NPY interneurons blocks mechanical and thermal hypersensitivity in models of inflammatory and neuropathic pain

We next assessed the effects of chemogenetically activating NPY-INs in the context of inflammatory and neuropathic pain (*Figure 4A, D, G*). Intraplantar complete Freund's adjuvant (CFA) resulted in punctate mechanical and heat hypersensitivity of the ipsilateral paw of AAV.flex.hM3Dq-mCherry-injected Npy-Cre mice that received i.p. injection of vehicle prior to behavioural testing. However, the mechanical and heat hypersensitivity were completely blocked in mice treated with CNO, and the heat latencies were significantly increased above the pre-CFA baseline values (*Figure 4B, C*). Vehicle-treated AAV.flex.hM3Dq-mCherry-injected Npy-Cre mice that had undergone spared nerve injury (SNI) also displayed mechanical and heat hypersensitivity of the ipsilateral paw, compared to pre-surgery thresholds. Both the mechanical and heat hypersensitivity were blocked in CNO-treated mice (*Figure 4E, F*). Because de novo expression of NPY is known to occur in injured A-fibre afferents following nerve injury (*Honore et al., 2000*; *Intondi et al., 2010*; *Wakisaka et al., 1991*; *Wakisaka et al., 1992*), this could result in expression of hM3Dq in these afferents, thus confounding interpretation of our results. We therefore quantified the number of mCherry-labelled cells in the somatotopically relevant L4 and L5 DRG 4 weeks following SNI surgery in four mice (*Figure 4—figure supplement 1*). As expected, no labelled cells were observed contralateral to the AAV injection and SNI surgery in either DRG in any of the mice. A few mCherry-labelled cells were observed in both L4 and L5 DRG on the ipsilateral side (cells per DRG: L4 = 12.5 ± 2.3, L5 = 17.5 ± 3.6; *Figure 4—figure supplement 1*). Because the numbers of A-fibre sensory neurons within the mid-lumbar DRG are estimated to be in the thousands in mice (*Duchen and Scaravilli, 1977*; *Lawson, 1979*), it is highly unlikely that CNO-mediated activation of the very few hM3Dq-expressing cells observed in the L4 and L5 DRG following SNI would contribute to the blockade of neuropathic pain that we observed. We therefore conclude that this effect is due to activation of spinal inhibitory NPY-INs. We also assessed mCherry expression in the L4 and L5 DRG of five CFA-treated AAV.flex.hM3Dq-mCherry-injected Npy-Cre mice, 3 days following CFA injection. In contrast to nerve injury, neuropeptide upregulation is not observed in rodent DRG under inflammatory conditions (*Honore et al., 2000*; *Wakisaka et al., 1992*). As expected, we observed no mCherry-labelled cells in the contra- or ipsilateral L4 or L5 DRG of these mice (data not shown).

Spinal NPY signalling has been implicated in the suppression of neuropathic pain through inhibition of NPY Y1 receptor (Y1R)-expressing excitatory interneurons in the dorsal horn (*Solway et al., 2011*; *Nelson et al., 2019*; *Nelson et al., 2022*). Therefore, the suppression of neuropathic hypersensitivity that we observed during chemogenetic activation of NPY-INs could be due to GABAergic transmission, NPY signalling, or a combination of both. To assess the potential role of Y1R signalling, we systemically co-administered CNO and the Y1R-selective antagonist BMS 193885 (*Acton et al., 2019*) prior to behavioural testing in AAV.flex.hM3Dq-mCherry-injected Npy-Cre mice that had undergone SNI surgery. Administration of the Y1R antagonist did not alter the CNO-mediated suppression of tactile and heat hypersensitivity in these mice (*Figure 4E, F*), suggesting that action of NPY on Y1 receptors is not required for this effect.

In addition to evoked hypersensitivity, peripheral nerve injury induces ongoing neuropathic pain in rodents, as well as engaging affective–emotional responses to pain (*King et al., 2009*). To determine the contribution of NPY-INs to ongoing pain we tested whether CNO induced conditioned place preference (CPP) in a separate cohort of AAV.flex.hM3Dq-mCherry-injected Npy-Cre mice following SNI surgery (*Figure 4G*). A wild-type control group that had undergone SNI was also included to test for any possible preference of (or aversion to) the effects of CNO that could have resulted from off-target effects independent of DREADD activation. CNO did not induce preference or aversion in either of these experimental groups (*Figure 4H, I*, *Figure 4—figure supplement 2A, B*). However, using the same experimental setup we observed preference for a chamber paired with gabapentin in mice that had undergone SNI (*Figure 4—figure supplement 2C, D*), showing that the CPP method was sufficiently sensitive to detect ongoing neuropathic pain. Together, these findings suggest that activating NPY-INs may not alleviate ongoing pain in the SNI model. We also assessed SNI-induced cold hypersensitivity in the cohort of AAV.flex.hM3Dq-mCherry-injected Npy-Cre mice that were used

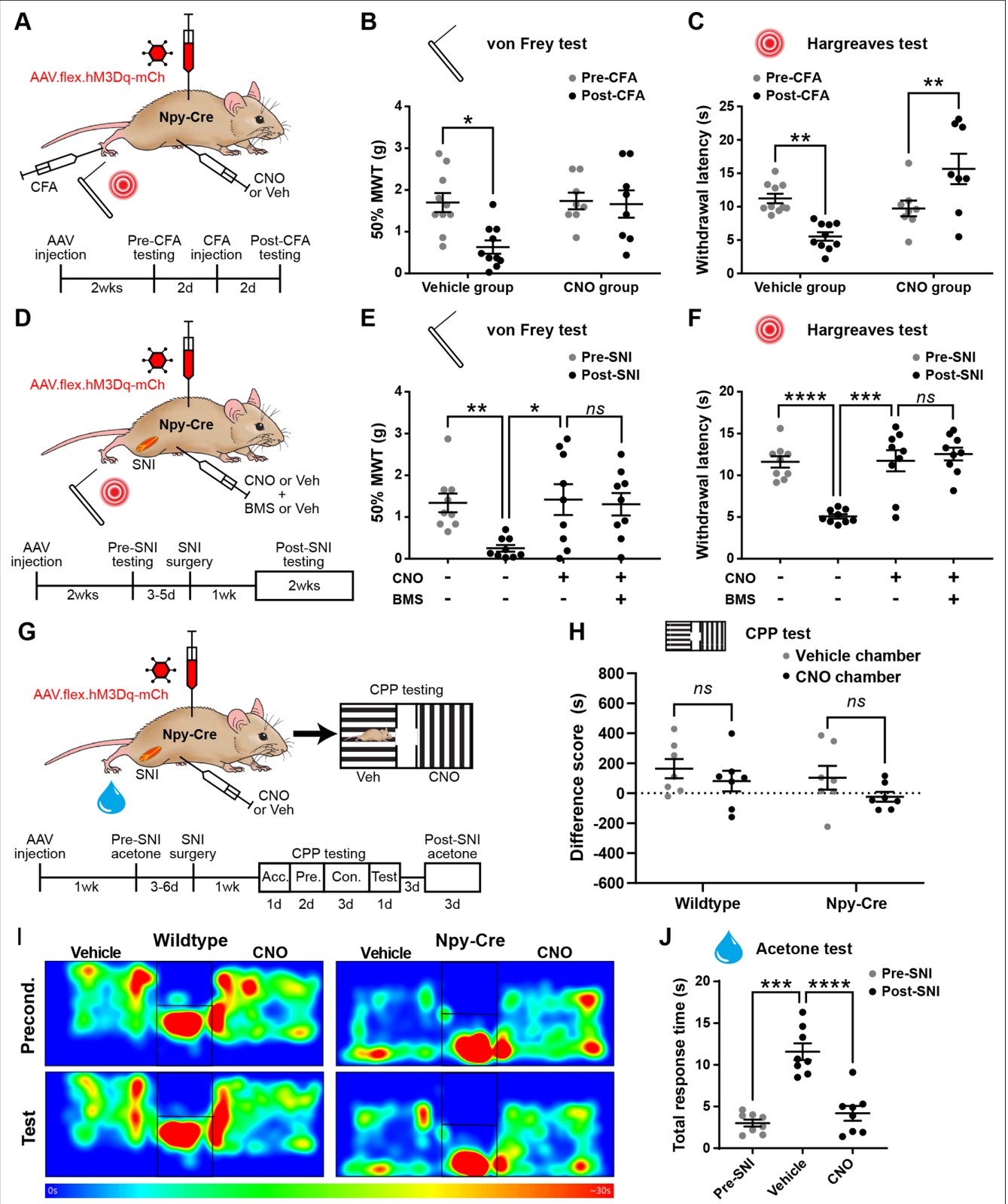

**Figure 4.** Activation of inhibitory neuropeptide Y (NPY) interneurons blocks mechanical and thermal hypersensitivity in models of inflammatory and neuropathic pain. (**A**) The experimental approach taken to generate the data presented in (**B, C**). (**B, C**) Vehicle control-treated AAV.flex.hM3Dq-mCherry spinal-injected Npy-Cre mice display marked reductions in mechanical withdrawal threshold (MWT) (**B**) and withdrawal latency to radiant heat (**C**) of the ipsilateral paw 2 days after intraplantar injection of complete Freund's adjuvant (CFA). Both mechanical and thermal hypersensitivity are

*Figure 4 continued on next page*

*Figure 4 continued*

blocked in clozapine-*N*-oxide (CNO)-treated mice (*n* = 10 vehicle group, 8 CNO group). (**D**) The experimental approach taken for the data presented in (**E, F**). Drug treatments were administered using a crossover design (*n* = 9). (**E, F**) Marked reductions in MWT (**E**) and withdrawal latency to radiant heat (**F**) are observed following spared nerve injury (SNI). These are blocked by CNO treatment, and this blockade persists in the presence of the Y1 antagonist BMS 193885 (BMS). (**G**) The experimental approach taken for the data presented in (**H–J**). Acc. = acclimation; Pre. = pre-conditioning; Con. = conditioning. (**H**) Neither wild-type control nor AAV.flex.hM3Dq-mCherry spinal-injected Npy-Cre mice displayed a conditioned place preference (CPP) to CNO following SNI (*n* = 7). (**I**) Heat maps of a representative mouse from each group demonstrating position and time spent in each chamber during preconditioning and post-conditioning test days. (**J**) A marked increase in the time spent responding to application of acetone to the ipsilateral paw (shaking, lifting, and/or licking) is seen in vehicle-treated AAV.flex.hM3Dq-mCherry spinal-injected Npy-Cre mice following SNI. This cold hypersensitivity is blocked when the same mice are treated with CNO (*n* = 8). Data are shown as individual values with mean ± standard error of the mean (SEM). *$p$ < 0.05, **$p$ < 0.01, ***$p$ < 0.001, ****$p$ < 0.0001; repeated-measures two-way ANOVA with Šidák's post-test in (**B, C, H**), repeated-measures one-way ANOVA with Šidák's post-test in (**E, F, J**).

The online version of this article includes the following figure supplement(s) for figure 4:

**Figure supplement 1.** Spared nerve injury (SNI) results in minimal ipsilateral hM3Dq expression in L4/5 DRG of AAV.flex.hM3Dq-mCherry spinal-injected Npy-Cre mice.

**Figure supplement 2.** Gabapentin administration, but not chemogenetic activation of NPY-INs, induces conditioned place preference (CPP) following spared nerve injury (SNI).

for CPP testing (*Figure 4G*). We observed a marked increase in the duration of the response to an acetone droplet applied to the ipsilateral hindpaw relative to the pre-SNI baseline when the mice had been dosed with a vehicle control. Administration of CNO completely blocked this hypersensitivity (*Figure 4J*), demonstrating a reversal of SNI-induced cold allodynia when NPY-INs are activated. This result also demonstrates that the lack of CPP in the chemogenetic experiments was not due to a failure to activate NPY-INs.

In summary, chemogenetic activation of NPY-INs supresses both mechanical and thermal hypersensitivity in models of inflammatory and neuropathic pain, and the suppression of neuropathic hypersensitivity appears to be mediated predominantly by GABAergic transmission from NPY-INs. However, NPY-IN activation does not appear to affect ongoing pain in the neuropathic model.

## Toxin-mediated silencing of NPY interneurons causes spontaneous itch and enhances pruritogen-evoked itch but does not alter nocifensive reflexes

We then tested whether tetanus toxin light chain (TeLC)-mediated silencing of NPY-INs following spinal injection of AAV.flex.TeLC.eGFP into Npy-Cre mice altered pain- or itch-related behaviours (*Figure 5A* and *Figure 5—figure supplement 1A*). Immunohistochemical assessment of the overlap of NPY and GFP expression in these mice demonstrated a very similar specificity and efficacy of expression in NPY-INs to that described above for other viral constructs. In animals injected with AAV.flex.TeLC.eGFP 82.6% ± 5.0% of GFP-positive cells co-expressed NPY and 61.7% ± 3.5% of NPY-positive cells co-expressed GFP (*Figure 5—figure supplement 1F, G*).

Compared to AAV.flex.eGFP-injected controls, AAV.flex.TeLC.eGFP-injected Npy-Cre mice displayed significant enhancement of CQ-induced itch when tested 4–6 days after AAV injection ($p$ <0.0001, two-way ANOVA with Tukey's post-test, *Figure 5B*). Approximately two-thirds of AAV.flex.TeLC.eGFP-injected mice also developed skin lesions on the ipsilateral hindlimb, within the corresponding dermatomes, by day 7 (*Figure 5D, E*). This phenotype was never observed in the AAV.flex.eGFP-injected controls, and strongly suggests development of spontaneous itch following silencing of NPY-INs. Consistent with this interpretation, we observed a significant increase in the time spent biting the calf prior to CQ injection in AAV.flex.TeLC.eGFP-injected Npy-Cre mice compared to AAV.flex.eGFP-injected mice of the same genotype ($p$ = 0.0014, two-way ANOVA with Tukey's post-test, *Figure 5C*). In contrast to these marked effects on itch-related behaviours, silencing of NPY-INs did not significantly alter punctate tactile or thermal nocifensive thresholds at 4–6 days after AAV injection (*Figure 5—figure supplement 1A–D*). Motor co-ordination, as assessed by rotarod, was also unaffected by NPY-IN silencing; however, a small but significant improvement was detected in AAV.flex.eGFP-injected mice relative to their baseline pre-surgery performance, most likely reflecting a mild training effect (*Figure 5—figure supplement 1E*). Taken together these findings indicate that tonic activity of NPY-INs suppresses itch, but has no obvious impact on nociceptive thresholds.

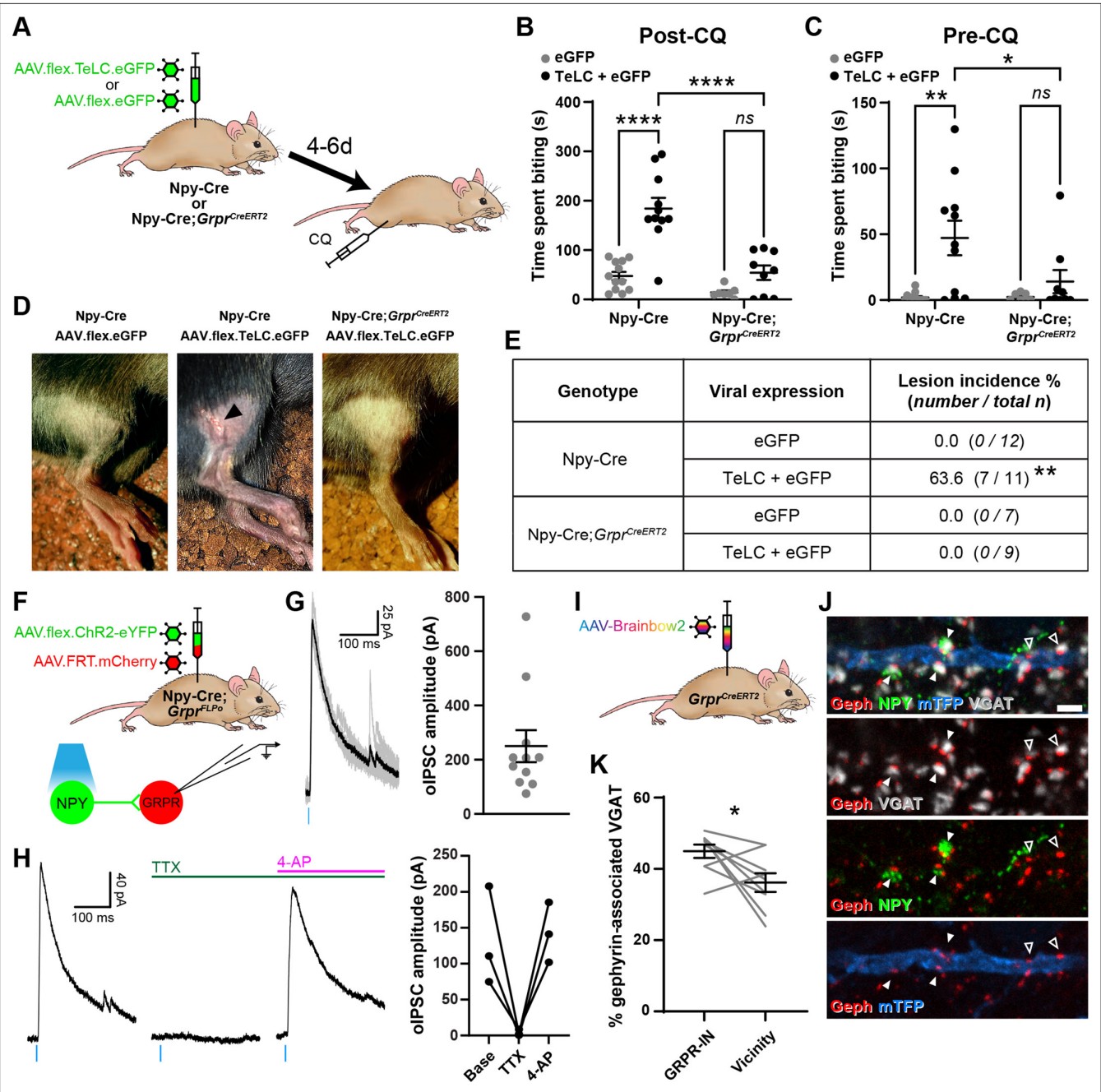

**Figure 5.** Increased itch caused by silencing neuropeptide Y (NPY) interneurons operates through a circuit involving GRPR-expressing excitatory interneurons. (**A**) The experimental approach used to generate the data presented in (**B–E**). (**B**) Silencing of NPY-INs by viral expression of tetanus toxin light chain (TeLC) in AAV.flex.TeLC.eGFP spinal-injected Npy-Cre mice results in a significant enhancement of chloroquine-evoked itch (Post-CQ), compared to that seen in AAV.flex.eGFP-injected controls. This enhancement of CQ-evoked itch is significantly reduced when NPY- and GRPR-INs are simultaneously silenced by injecting AAV.flex.TeLC.eGFP into Npy-Cre;*Grpr^CreERT2^* mice. The numbers of mice per group are as outlined in the table in (**E**). (**C**) Silencing of NPY-INs by TeLC also results in the development of spontaneous itch behaviour as assessed over 30 min prior to CQ administration (Pre-CQ). This spontaneous itch is also significantly reduced when GRPR-INs are simultaneously silenced. (**D**) Representative images of a skin lesion on the calf of an AAV.flex.TeLC.eGFP-injected Npy-Cre mouse (arrowhead, middle image), and the lack of lesions in AAV.flex.eGFP-injected Npy-Cre or AAV.flex.TeLC.eGFP-injected Npy-Cre;*Grpr^CreERT2^* mice. (**E**) Table outlining the incidence of lesions in AAV.flex.eGFP- or AAV.flex.TeLC.eGFP-injected Npy-Cre or Npy-Cre;*Grpr^CreERT2^* mice. Lesions were observed in approximately two-thirds of AAV.flex.TeLC.eGFP-injected Npy-Cre mice, but never in AAV.flex.TeLC.eGFP-injected Npy-Cre;*Grpr^CreERT2^* mice, nor in AAV.flex.eGFP-injected control groups. (**F**) The experimental approach used to generate the data presented in (**G, H**). (**G**) Optogenetic activation of NPY-INs induces monosynaptic optogenetically evoked IPSCs (oIPSCs) in GRPR-INs. Representative traces of oIPSCs recorded in a GRPR neuron are shown on the left, with six individual oIPSCs in grey and an averaged trace in black. Quantification of

*Figure 5 continued on next page*

*Figure 5 continued*

the mean peak amplitude of oIPSCs recorded in 11 GRPR-INs is shown on the right. For all 11 cases, all 6 light stimuli resulted in oIPSCs with no failures. (**H**) Example traces from a GRPR-IN (left) show that oIPSCs are blocked by tetrodotoxin (TTX) and reinstated by 4-aminopyridine (4-AP); quantification of mean peak oISPC amplitude from three GRPR-INs is shown on the right. (**I**) The experimental approach used to generate the data presented in (**J, K**). (**J**) Filled arrowheads mark three examples of NPY-immunoreactive (green) inhibitory boutons synapsing onto dendrite of a Brainbow-labelled GRPR-IN (mTFP, blue). Inhibitory synapses were defined as VGAT-positive profiles (grey) in contact with gephyrin puncta (red). Open arrowheads mark two examples of NPY-negative inhibitory synapses on the Brainbow-labelled dendrite. Scale bar = 2 µm. (**K**) Quantification of the percentage of inhibitory synapses on to nine GRPR-INs (n = 3 mice), or in the vicinity of those cells, at which the presynaptic VGAT bouton is NPY-immunoreactive. Data are shown as individual values with mean ± standard error of the mean (SEM) in (**B, C, G**), individual values in (**H**) and individual matched values with mean ± SEM in (**K**). *p < 0.05, **p < 0.01, ****p < 0.0001; two-way ANOVA with Tukey's post-test in (**B, C**), Fisher's exact test with Bonferroni correction in (**E**) and Wilcoxon matched pairs test in (**K**).

The online version of this article includes the following figure supplement(s) for figure 5:

**Figure supplement 1.** Silencing of NPY-INs does not affect nociceptive thresholds.

**Figure supplement 2.** NPY-INs generate GABAergic and glycinergic inhibition of GRPR-INs.

## Increased itch caused by silencing NPY interneurons operates through a circuit involving GRPR-expressing excitatory interneurons

Several studies have shown that GRPR-expressing excitatory dorsal horn interneurons (GRPR-INs) are crucial for pruritogen-evoked itch (*Mishra and Hoon, 2013*; *Sun and Chen, 2007*; *Sun et al., 2009*), while it has been proposed that they are not required for mechanical itch (*Bourane et al., 2015*; *Acton et al., 2019*; but see *Chen et al., 2020*). To assess whether signalling through GRPR-INs was required for the itch-related behaviours that we observed when NPY-INs were silenced, we crossed Npy-Cre and *Grpr^{CreERT2}* mice and concomitantly silenced NPY-INs and GRPR-INs through spinal injection of AAV.flex.TeLC.eGFP (*Figure 5A*). AAV.flex.eGFP-injected mice of the same genotype were again used as a control group. Npy-Cre;*Grpr^{CreERT2}* mice that received injections of AAV.flex.TeLC.eGFP showed no significant difference in CQ-induced itch, compared to AAV.flex.eGFP-injected controls (p = 0.34, two-way ANOVA with Tukey's post-test, *Figure 5B*). However, when comparing Npy-Cre and Npy-Cre;*Grpr^{CreERT2}* mice that had received injections of AAV.flex.TeLC.eGFP, we found that the Npy-Cre;*Grpr^{CreERT2}* mice showed significantly less CQ-induced itch behaviour than Npy-Cre mice (p < 0.0001, two-way ANOVA with Tukey's post-test, *Figure 5B*). Furthermore, AAV.flex.TeLC.eGFP-injected Npy-Cre;*Grpr^{CreERT2}* mice did not display a significant increase in spontaneous biting prior to CQ administration (compared to AAV.flex.eGFP-injected controls; p = 0.82, two-way ANOVA with Tukey's post-test, *Figure 5C*) and never developed skin lesions (*Figure 5D, E*). These data demonstrate that both the spontaneous itch and the increased pruritogen-evoked itch observed following silencing of NPY-INs are at least partly transmitted via GRPR-INs.

This led us to ask whether NPY-INs provide direct inhibitory synaptic input to GRPR-INs. To investigate this we performed ex vivo patch clamp experiments in spinal cord slices from Npy-Cre;*Grpr^{FLPo}* mice that had received intraspinal injections of AAV.flex.ChR2-eYFP together with AAV.FRT.mCherry, resulting in expression of eYFP-tagged channelrhodopsin in NPY-INs and mCherry in GRPR-INs (*Figure 5F*). Recordings were made from 11 mCherry+ cells and all of these exhibited an oIPSC (with no failures) when the slice was illuminated with brief pulses of blue light, with a mean peak oIPSC amplitude of 250.1 ± 58.9 pA (*Figure 5G*). In 3/3 of these cells oIPSCs were abolished by the application of tetrodotoxin (TTX) and rescued by the addition of 4-AP (*Figure 5H*), confirming that they were monosynaptic, and therefore that NPY-INs directly inhibit GRPR-INs. The GABAergic/glycinergic nature of the oIPSCs was assessed in three of the cells (*Figure 5—figure supplement 2A–C*). In 2/3 cells the oIPSC was gabazine sensitive/strychnine insensitive, indicating GABA-mediated inhibition, while in the other cell the oIPSC was sensitive to both gabazine and strychnine, indicating mixed GABA and glycine inhibition (*Figure 5—figure supplement 2B, C*). Although *Acton et al., 2019* provided evidence that GRPR-INs lack the Y1R, *Chen et al., 2020* reported that 35% of GRPR cells had *Y1R* mRNA. We therefore tested the effect of bath-applying the Y1R agonist [Leu^{31},Pro^{34}]-neuropeptide Y, while recording from GRPR-INs (*Figure 5—figure supplement 2D*). We found that all 10 cells tested failed to show an outward current in response to [Leu^{31},Pro^{34}]-neuropeptide Y (*Figure 5—figure supplement 2E, F*), suggesting that NPY acting on the Y1R is unlikely to have made a significant contribution to the suppression of GRPR cells by NPY-INs. We also investigated inhibitory NPY-IN input to GRPR-INs anatomically. To do this, we quantified the proportion of inhibitory synaptic contacts onto

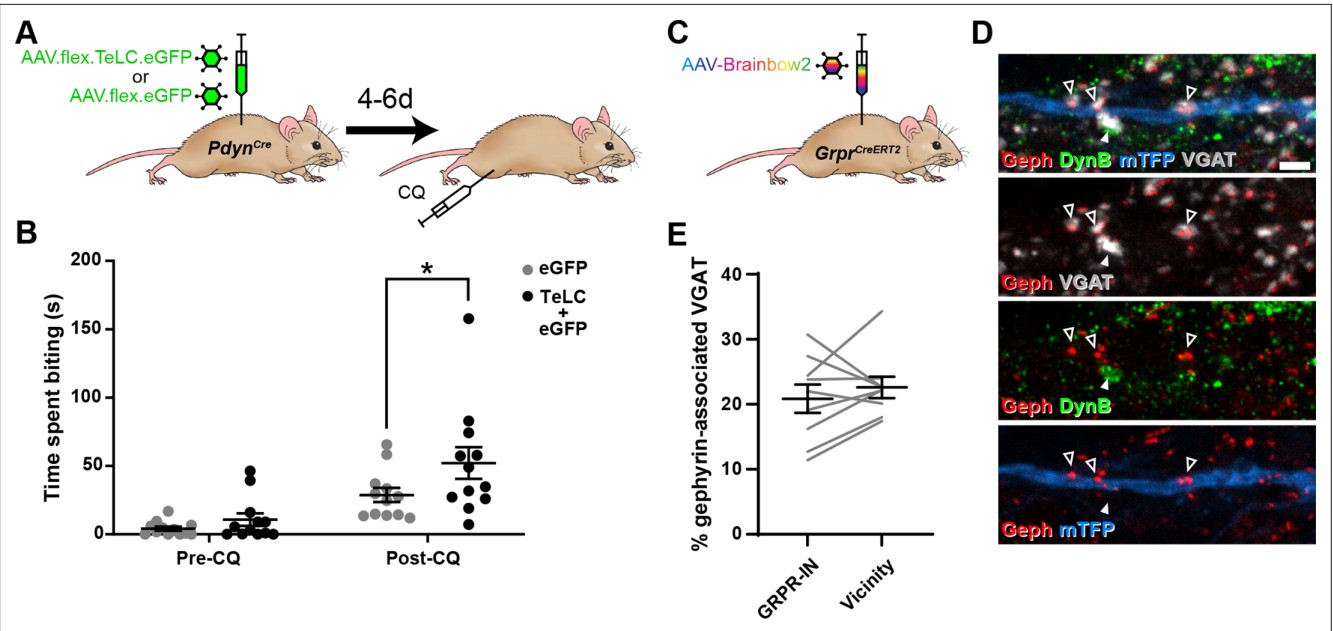

**Figure 6.** Toxin-mediated silencing of dynorphin-expressing inhibitory interneurons enhances pruritogen-evoked itch. (**A**) The experimental approach used to generate the data presented in (**B**). (**B**) Silencing of Dyn-INs by viral expression of tetanus toxin light chain (TeLC) in *Pdyn^Cre^* mice results in a significant enhancement of chloroquine-evoked itch (Post-CQ), compared to eGFP-expressing controls (p = 0.0379; repeated-measures two-way ANOVA with Šidák's post-test). No significant difference was observed in biting time prior to injection with chloroquine (Pre-CQ; p = 0.7431; *n* = 12 for both GFP and TeLC groups). (**C**) The experimental approach used to generate the data presented in (**D, E**). (**D**) Filled arrowhead marks an example of a DynB-expressing (green) inhibitory bouton synapsing onto dendrite of a Brainbow-labelled GRPR-IN (mTFP, blue). Inhibitory synapses were defined as VGAT-positive profiles (grey) in contact with gephyrin puncta (red). Open arrowheads mark three examples of DynB-negative inhibitory synapses on the Brainbow-labelled dendrite. Scale bar = 2 μm. (**E**) Quantification of the percentage of total inhibitory synapses on to nine GRPR-INs (n = 3 mice), or in the vicinity of those cells, that contain DynB. No significant difference was observed between these proportions (p = 0.3438; Wilcoxon matched pairs test). Data are shown as individual values with mean ± standard error of the mean (SEM) in (**B**) and individual matched values with mean ± SEM in (**E**).

The online version of this article includes the following figure supplement(s) for figure 6:

**Figure supplement 1.** Injection of AAV.flex.TeLC.eGFP does not cause exaggerated itch in wild-type mice.

GRPR-INs (labelled through spinal injection of Cre-dependent AAV-Brainbow2 into *Grpr^CreERT2^* mice; *Figure 5I*) at which NPY was present in the presynaptic bouton. Inhibitory synapses were identified by the presence of VGAT-positive presynaptic boutons apposed to puncta of the postsynaptic protein gephyrin. Many of the gephyrin puncta on the GRPR-INs were contacted by NPY-IR boutons, and these accounted for 45.0% ± 1.9% of all inhibitory synapses on the GRPR-INs (*Figure 5J, K*). This was significantly higher than the proportion of inhibitory (VGAT+) boutons in the vicinity of the analysed cells that contained NPY (36.2% ± 2.6%; *Figure 5K*). Together these data provide strong evidence that NPY-INs selectively target GRPR-INs and generate a powerful GABAergic inhibition of these cells.

## Toxin-mediated silencing of dynorphin interneurons enhances pruritogen-evoked itch

Dorsal horn inhibitory interneurons that co-express dynorphin and galanin have been implicated in suppression of itch as their activation reduces pruritogen-evoked itch (*Huang et al., 2018*; *Liu et al., 2019*), while constitutive loss of B5-I neurons, which include this population, results in enhanced pruritogen-evoked itch and skin lesions due to spontaneous scratching (*Kardon et al., 2014*; *Ross et al., 2010*). In addition, *Brewer et al., 2020* reported that chemogenetic inhibition of dynorphin lineage cells increases pruritogen-evoked itch. Given the well-established role of dynorphin-expressing interneurons (Dyn-INs) in suppressing itch, we compared the effect of silencing these cells with that of silencing the NPY-INs, in order to explore a potential overlap of function. Although dynorphin is also expressed by a subset of dorsal horn excitatory interneurons, we have shown that these are largely restricted to areas innervated by afferents from glabrous skin (*Huang et al., 2018*). We injected AAV.flex.TeLC.eGFP (or AAV.flex.eGFP as a control) into the dorsal horn of the L3 segment of *Pdyn^Cre^* mice

for these experiments (*Figure 6A*). This segment was chosen for two reasons: (1) it receives input from the region of calf that we used to test the effect of pruritogens, and (2) it receives input exclusively from hairy skin, and therefore the great majority of virally transfected Dyn-INs are likely to be inhibitory cells (those that co-express dynorphin and galanin) (*Huang et al., 2018*).

In contrast to the effects of silencing NPY-INs, silencing of Dyn-INs never resulted in skin lesions, and the time spent biting the calf prior to CQ injection appeared to be unaffected (*Figure 6B*; p = 0.7431, two-way repeated measures ANOVA with Šidák's post-test). This suggests that silencing Dyn-INs does not result in spontaneous itch. Silencing of Dyn-INs did result in enhanced CQ-evoked itch, when compared to AAV.flex.eGFP-injected controls (*Figure 6B*; p = 0.0379, two-way repeated measures ANOVA with Šidák's post-test). However, this enhancement was markedly less pronounced than that observed following silencing of NPY-INs (compare *Figure 6B* with *Figure 5B*). To confirm that the effects of TeLC-mediated silencing resulted from targeting of Cre-expressing cells, we also injected either AAV.flex.TeLC.eGFP or AAV.flex.eGFP into the L3 segments of wild-type mice, and assessed CQ-evoked itch behaviours 4–6 days later (*Figure 6—figure supplement 1A*). As expected in these control animals, there was no significant difference in the time spent biting the calf between AAV.flex.TeLC.eGFP- or AAV.flex.eGFP-injected mice either before or after injection of CQ (*Figure 6—figure supplement 1B*).

These findings suggest that while both NPY- and Dyn-INs can suppress itch, the NPY population has a more substantial role in this mechanism. One explanation for this could be that although Dyn-INs form inhibitory synapses onto GRPR cells (*Liu et al., 2019*), the density of these synapses is less than that of those arising from the NPY-INs. To test this, we assessed contacts from inhibitory dynorphin cells onto GRPR-INs (*Figure 6C–E*), using an antibody against dynorphin B (DynB) and found that these constituted only 20.9% ± 2.2% of the inhibitory synapses on these cells. This did not differ significantly from the proportion of inhibitory boutons that contained DynB in the vicinity of the analysed cells (22.6% ± 1.6%; *Figure 6D, E*). Collectively, these results suggest that unlike NPY-INs, Dyn-INs do not preferentially target GRPR-INs. In addition, they contribute a far lower proportion of inhibitory synapses on the GRPR-INs (compared to the NPY-INs), and loss of this input has a much less dramatic effect on itch.

## Discussion

Inhibitory interneurons in the SDH play an important role in suppressing pain and itch. NPY-expressing cells constitute around a third of the inhibitory neurons in this region and are also present in deeper laminae. Previous studies in which NPY-lineage neurons were ablated demonstrated that these cells are responsible for preventing mechanical itch through a mechanism involving NPY and the Y1 receptor (*Bourane et al., 2015*; *Acton et al., 2019*). Here we show, by selectively activating those cells that continue to express the peptide, that the NPY cells inhibit acute nocifensive reflexes and reduce mechanical and thermal hypersensitivity in both inflammatory and neuropathic pain models. In addition, they strongly suppress itch evoked by CQ. Silencing the NPY cells causes spontaneous itch and exaggerated responses to CQ, and both of these effects are reduced by simultaneously silencing GRPR-expressing excitatory interneurons, indicating that suppression of itch by the NPY cells operates through downstream GRPR-INs.

### A broad inhibitory role for NPY cells

Our findings indicate that NPY-INs have a far broader role in suppressing pain- and itch-related behaviours than had been suggested by previous studies that used the same Npy-Cre line (*Bourane et al., 2015*; *Pan et al., 2019*; *Acton et al., 2019*), despite the fact that we were targeting a more restricted neuronal population. The differences in experimental findings are likely to result from two methodological issues: (1) the technique used to target cells, and (2) the use of loss-of-function or gain-of-function approaches. In each of these other studies, cells were targeted by an intersectional genetic approach that limited expression to spinal cord and brainstem, but would have included a large additional group of inhibitory neurons that expressed NPY only during development (*Bourane et al., 2015*). Here, we used an alternative strategy to restrict expression: intraspinal injection of AAVs coding for Cre-dependent constructs (*Foster et al., 2015*; *Huang et al., 2018*). While this approach failed to capture a minority of NPY-expressing neurons, it enabled us to target a large number of

these cells. Importantly, expression was restricted to those cells that continue to express NPY. This was confirmed by our finding that up to 85% of the virally transfected cells contained detectable levels of NPY.

The main differences in interpreting the roles of NPY cells are likely to depend on whether the cells were inactivated (through ablation or synaptic silencing) or chemogenetically activated. In agreement with *Bourane et al., 2015*, we found that silencing NPY cells had no effect on acute nociceptive thresholds. However, chemogenetically activating these cells increased thresholds for both thermal and mechanical nocifensive reflexes, and reduced hypersensitivity in neuropathic and inflammatory pain models. Interestingly, *Acton et al., 2019* also observed an anti-nociceptive effect on mechanical stimuli when they chemogenetically activated NPY-lineage neurons, but attributed this to ectopic activation of Y1 receptors on primary sensory neurons. However, although Y1 is present in cell bodies of some primary sensory cells, it is not thought to traffic to their central terminals (*Zhang et al., 1994*; *Nelson and Taylor, 2021*). *Acton et al., 2019* did not test whether activating NPY-lineage neurons had any effect on responses to thermal stimuli, or on neuropathic/inflammatory hypersensitivity, so it is not possible to compare our findings in these contexts. The most likely explanation for discrepancies between the findings of loss-of-function and gain-of-function studies is that although NPY cells have an antinociceptive action, other interneurons provide sufficient inhibition to maintain nocifensive reflexes when NPY cells are silenced. Our findings therefore indicate that NPY-INs have a far broader role in somatosensory processing than was previously recognised.

## NPY cells suppress spontaneous and pruritogen-evoked itch

Our previous studies (*Huang et al., 2018*; *Kardon et al., 2014*) had implicated dynorphin/galanin cells in suppression of pruritogen-evoked itch. This was based on the findings that $Bhlhb5^{-/-}$ mice (which lack these cells) show exaggerated responses to pruritogens (*Kardon et al., 2014*), and that chemogenetic activation of Dyn-INs suppressed CQ-evoked itch (*Huang et al., 2018*). In support of this, *Liu et al., 2019* subsequently showed that activating galanin-expressing cells also suppresses pruritogen-evoked itch. This anti-pruritic action is likely to involve dynorphin acting on κ-opioid receptors (*Kardon et al., 2014*) as well as direct inhibition of GRPR cells (which are an integral part of the spinal itch pathway) by GABA and/or glycine released from the dynorphin/galanin cells (*Liu et al., 2019*). Liu et al also showed that ablating galanin-expressing cells enhances pruritogen-evoked itch, and consistent with this we find enhancement of CQ-evoked itch when cells belonging to this population are silenced by injecting AAV.flex.TeLC into $Pdyn^{Cre}$ mice.

Here, we show that activating NPY cells also strongly suppresses CQ-evoked itch. This is at odds with findings of *Acton et al., 2019*, who reported that chemogenetic activation of NPY-lineage neurons failed to alter scratching in response to CQ. There are technical differences between these studies, since Acton et al. used a reporter mouse line to express hM3Dq, and injected CQ intradermally behind the ear. The discrepancy between the results of these studies is most likely to result from higher levels of DREADD expression following viral transfection, and therefore more effective neuronal activation. However, there may also have been a contribution from regional differences in the itch tests used (hindlimb vs. head), as well as in the neuronal populations targeted (as noted above). Although *Bourane et al., 2015* reported that ablating ~70% of NPY-lineage neurons had no effect on itch evoked by CQ, we found that synaptic silencing of the NPY cells with TeLC increased CQ-evoked itch, and often resulted in development of skin lesions, presumably secondary to the spontaneous itch-related biting that was also observed. In fact, the antipruritic action of the NPY-INs may be more powerful than that of the dynorphin/galanin cells, since TeLC silencing in the $Pdyn^{Cre}$ mouse caused less of an increase in CQ-evoked itch behaviour (compared to silencing in the Npy-Cre line) and did not result in the development of spontaneous itch or associated skin lesions. Nonetheless, our findings demonstrate that both NPY-INs and Dyn-INs contribute to the suppression of pruritogen-evoked itch thus revealing an overlap of function of these neurochemically distinct inhibitory interneuron populations.

## NPY cells operate through a circuit involving GRPR neurons

Both spontaneous and CQ-evoked itch behaviours were suppressed when GRPR and NPY cells were silenced simultaneously, and we show directly, using both anatomical and electrophysiological methods, that the NPY cells provide a strong inhibitory input to GRPR-INs. This indicates

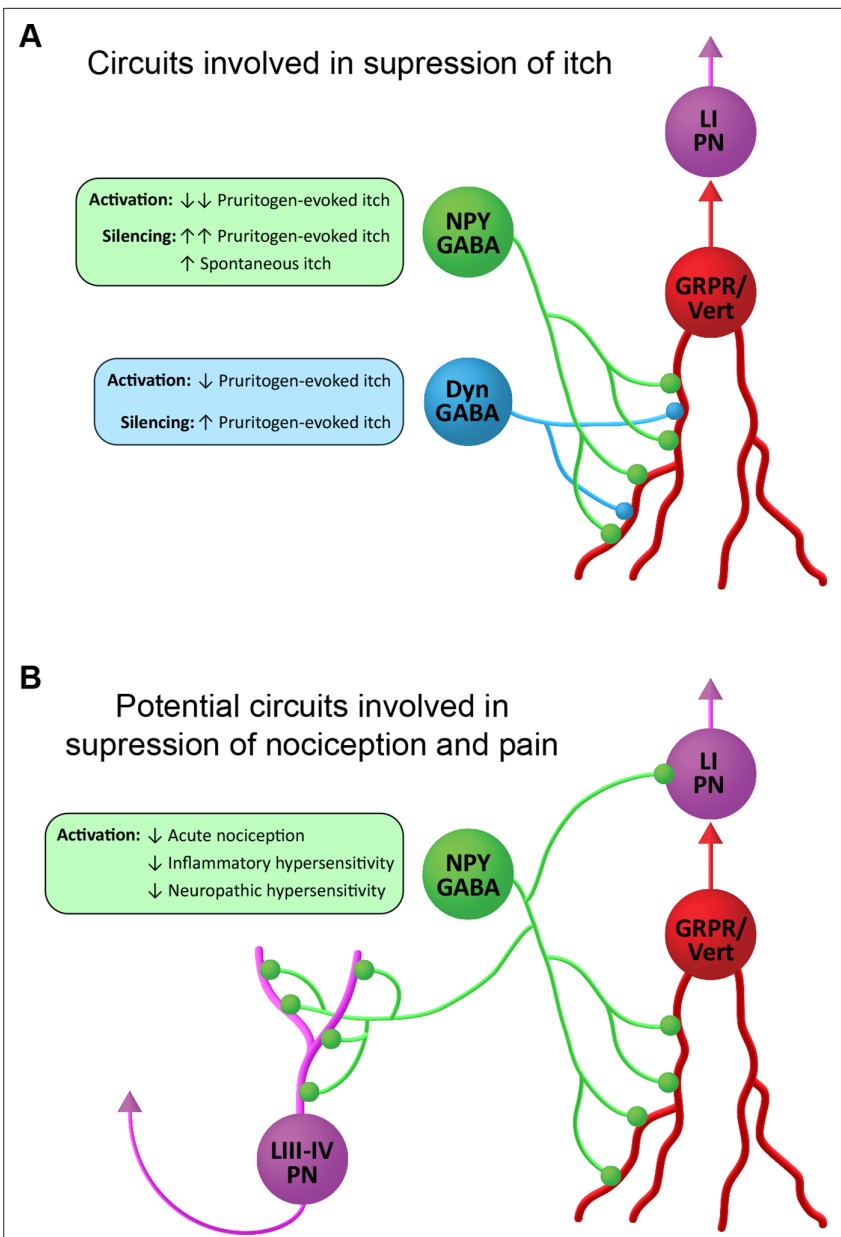

**Figure 7.** Suggested roles of neuropeptide Y (NPY)-expressing inhibitory interneurons in spinal itch and pain circuits. (**A**) Circuits involved in suppression of itch. NPY-INs provide a high proportion (45%) of the inhibitory synapses on GRPR-expressing excitatory interneurons. The GRPR cells have vertical (Vert) morphology and are thought to transmit itch- and pain-related information to spinal projection neurons in lamina I (LI PNs). Pruritogen-evoked itch is markedly reduced by NPY-IN activation, while silencing of NPY-INs enhances pruritogen-evoked itch and results in spontaneous itch. Dyn-INs also provide inhibitory input to GRPR cells, but only account for ~20% of their inhibitory synapses. Silencing of Dyn-INs increases pruritogen-evoked itch, but to a lesser degree than NPY-IN silencing, and without the appearance of spontaneous itch. Activation of Dyn-INs has previously been shown to reduce itch in response to a range of pruritogens (see *Huang et al., 2018*). (**B**) Potential circuits involved in suppression of nociception and pain. Vertical cells are thought to be involved in the transmission of both normal and pathological pain signals through their input to LI PNs. Chemogenetic activation of GRPR-expressing vertical cells elicits both itch- and pain-related behaviours (*Polgár et al., 2023*). NPY-INs may therefore act to supress acute nociception, as well as inflammatory and neuropathic hypersensitivity, via inhibition of vertical cells. Additionally, NPY-INs have previously been shown to directly innervate nociceptive projection neurons of the anterolateral system in lamina I and laminae III–IV (LIII–IV PN).

that GRPR cells are downstream of the NPY cells (**Figure 7A**). This inhibitory input to GRPR cells appears to be even more powerful than that originating from the dynorphin/galanin cells, since NPY-IR boutons accounted for 45% of the inhibitory synapses on the GRPR cells, compared to the 21% from dynorphin-IR boutons. Consistent with this we found that optogenetic activation of NPY cells elicited oIPSCs in all of the GRPR cells tested. Interestingly, these were of much higher mean amplitude (~250 pA), than the ~80 pA oIPSCs reported by **Liu et al., 2019** in GRPR cells when galanin cells were optogenetically activated using a very similar experimental approach. The inhibition of GRPR cells by NPY-INs is likely to be predominantly GABAergic, since oIPSCs were reduced by gabazine in all cells (with one also sensitive to strychnine). Also, consistent with previous evidence showing that the majority of GRPR cells lack Y1 receptors (**Acton et al., 2019**; **Chen et al., 2020**), we did not detect outward currents in any of the GRPR cells that were tested with a Y1 agonist.

Although GRPR-expressing excitatory interneurons have been strongly implicated in itch, we have recently shown that these cells respond to noxious as well as pruritic stimuli, that they correspond morphologically to a class of SDH excitatory interneurons known as vertical cells, and that chemogenetically activating them results in behaviours reflecting both pain and itch (**Polgár et al., 2023**). Vertical cells provide input to lamina I projection neurons (**Lu and Perl, 2005**), and are thought to form an integral part of circuits that underlie both normal and pathological pain (**Duan et al., 2014**; **Lu et al., 2013**; **Peirs and Seal, 2016**). It is already known that axons of NPY cells directly innervate lamina I projection cells, as well as a population of nociceptive projection neurons in laminae III–V of the dorsal horn (**Kókai et al., 2022**; **Polgár et al., 1999**). This direct input to ALS projection neurons will presumably contribute to the antinociceptive action of the NPY cells. The present findings raise the possibility that the powerful inhibitory GABAergic NPY-GRPR circuit that we have identified contributes not only to the alleviation of itch, but also to the suppression of nocifensive reflexes and the reduction of hypersensitivity in persistent pain states (**Figure 7B**).

## Activating NPY cells suppresses hypersensitivity in persistent pain states

Importantly, in addition to its effect on acute nocifensive reflexes and itch, activating NPY cells also blocked thermal and mechanical hypersensitivity in both inflammatory and neuropathic pain states. In the SNI model, we found that administration of a Y1 antagonist had no effect on the reversal of mechanical and heat hypersensitivity when NPY cells were activated. NPY acting on Y1 receptors expressed by spinal neurons is known to reduce signs of neuropathic pain (**Solway et al., 2011**; **Nelson et al., 2019**; **Intondi et al., 2008**); however, it appears that chemogenetic activation of NPY cells generated GABAergic inhibition that was sufficiently powerful to reverse the hypersensitivity independently of Y1 signalling. Interestingly, our CPP findings suggest that activating NPY neurons may not suppress on-going pain in the SNI model, implying that on-going and evoked components of neuropathic pain operate through different circuits at the spinal cord level.

Previous studies have tested the effects of chemogenetically activating other inhibitory interneuron populations on persistent pain states. Glycinergic cells account for the majority of inhibitory interneurons in deep dorsal horn and are largely separate from the NPY-INs (**Rowan et al., 1993**; **Miranda et al., 2022**). **Foster et al., 2015** showed that activating these cells reduced responses to acute thermal and mechanical noxious stimuli and suppressed mechanical hypersensitivity in the chronic constriction injury model. Activation of PV-expressing inhibitory interneurons reduced inflammatory and neuropathic mechanical allodynia, but had no effect on heat hypersensitivity (**Petitjean et al., 2015**). A recent study by **Albisetti et al., 2023** found that activating a population of dorsal horn inhibitory interneurons defined by expression of *Kcnip2* suppressed cold allodynia in a neuropathic model. However, our findings apparently provide the first evidence that activating dorsal horn inhibitory interneurons can suppress heat hypersensitivity, in addition to cold and mechanical allodynia, in persistent pain states. A population of NPY-expressing inhibitory interneurons with a similar laminar location has recently been identified in human spinal cord (**Yadav et al., 2022**). These cells therefore provide an attractive target for the treatment of neuropathic pain, particularly for the significant cohort of patients who experience thermal hyperalgesia (**Baron et al., 2017**).

# Materials and methods

## Experimental model and subject details

All experiments were approved by the Ethical Review Process Applications Panel of the University of Glasgow, and were carried out in accordance with the European Community directive 86/609/EC, the UK Animals (Scientific Procedures) Act 1986 and ARRIVE guidelines. The following transgenic mouse lines were used in this study: the GENSAT BAC transgenic RH26 Npy-Cre line, which express Cre recombinase under control of the NPY promoter (*Gerfen et al., 2013*); the Ai9 Cre reporter line, in which a loxP-flanked STOP cassette prevents CAG promoter-driven transcription of tdTomato; the *Grpr^CreERT2* line, in which 2A-linked optimised Cre recombinase fused with the ligand-binding domain of the estrogen receptor is inserted into the 3'UTR of the *Grpr* gene *Mu et al., 2017*; the *Grpr^FLPo* line, in which 2A-linked FlpO recombinase is fused with the last exon of the *Grpr* gene (*Liu et al., 2019*); and the *Pdyn^Cre* line, in which an IRES site fused to Cre recombinase is inserted downstream of the stop codon of the *Pdyn* gene (*Krashes et al., 2014*). Further details of these lines can be found in the Key Resources Table. Npy-Cre and Ai9, Npy-Cre and *Grpr^CreERT2*, Npy-Cre and *Grpr^FLPo*, or *Grpr^CreERT2* and Ai9 mice were crossed to produce Npy-Cre;Ai9, Npy-Cre;*Grpr^CreERT2*, Npy-Cre;*Grpr^FLPo*, and *Grpr^CreERT2*;Ai9 experimental animals, respectively. For experiments inolving the *Grpr^CreERT2* line, mice received 6 mg of tamoxifen (2× i.p. injections of 3 mg on consecutive days). For the Npy-Cre;*Grpr^CreERT2* mice, this was administered on the day of surgery and on the next day. Wild-type C57BL/6 mice were used for assessment of possible off-target effects of systemic CNO administration, and also to test for Cre-independent effects following injection of AAV.flex.TeLC.eGFP. Mice weighed 15–28 g and animals of both sexes were used, with care taken to include approximately equal numbers of males and females in each part of the study. The animals were between 5 and 14 weeks old at the time of tissue harvest (for anatomy), electrophysiological recording, or behavioural testing. Where drug treatment was given (except for CPP experiments), the treatment type, or the order in which mice received the drug or vehicle was randomised. For TeLC experiments, the viral construct used for each animal was randomised. In all of these cases, the experimenter was blind to the treatment or the viral construct used.

## Intraspinal AAV injections

Mice were anaesthetised with 1–2% isoflurane and placed in a stereotaxic frame. The skin was incised in the midline over the upper back and superficial muscle was removed from the vertebral column at the level of the T12 to L1 vertebrae, which were then clamped. The L3 and L5 spinal segments were injected through the T12/T13 and T13/L1 intervertebral spaces, respectively, whereas the L4 segment was injected via a hole drilled through the lamina of the T13 vertebra. Injections were performed by making a small slit in the dura and inserting a glass micropipette (outer/inner tip diameter: 60/40 μm) attached to a 10-μl Hamilton syringe, 400 μm lateral to the midline, and 300 μm below the pial surface. The following AAV constructs were used: AAV.flex.eGFP ($1.72 \times 10^9$ GC), AAV.flex.tdTomato ($1.76 \times 10^9$ GC), and AAV.flex.ChR2-eYFP ($5.09 \times 10^8$ GC), all from Penn Vector Core, PA, USA; AAV.flex.hM3Dq-mCherry, University of North Carolina Vector Core, NC, USA; or University of Zurich Viral Vector Facility, Switzerland ($3.8 \times 10^8$ or $7.65 \times 10^8$ GC, respectively); AAV.flex.eGFP ($2 \times 10^8$ GC), AAV.flex.TeLC.eGFP ($2 \times 10^8$ GC), and AAV.FRT.mCherry ($8.7 \times 10^8$ GC), all from University of Zurich Viral Vector Facility, Switzerland; and AAV-Brainbow2 ($1.5–5.96 \times 10^7$ GC), Addgene, MA, USA. Further details of the viruses used can be found in the Key Resources Table. 300 nl of virus was infused per injection site (or 500 nl for AAV-Brainbow2) at a rate of 30–40 nl/min using a syringe pump (Harvard Apparatus, MA, USA). Pipettes were left within the spinal cord for 5 min to minimise leakage of injectate. Once injections were complete the wound was closed and animals recovered with appropriate analgesic administration (0.3 mg/kg buprenorphine and 5 mg/kg carprofen). The success of spinal AAV injections was assessed by post hoc immunohistochemical staining for the appropriate fluorescent marker protein. Mice were only included for behavioural analyses if the AAV injection(s) into the spinal segments relevant to the dermatome(s) being tested were successful (L4 and L5 for plantar hindpaw-directed tests, L3 for calf skin-directed tests; see below). In some experiments Npy-Cre;*Grpr^CreERT2* mice were injected with AAV.flex.eGFP or AAV.flex.TeLC.eGFP, which should result in both NPY- and GRPR-expressing virus-infected cells being labelled with eGFP. This was confirmed for each injection site in each animal by two methods: (1) by assessing the distribution of eGFP-labelled cells, as NPY-INs are located throughout laminae I–III, whereas GRPR-INs are almost entirely restricted

to laminae I and IIo, and (2) by antibody co-staining for the inhibitory marker Pax2, as NPY-INs in laminae I–III are exclusively inhibitory, whereas GRPR-INs in laminae I and IIo are exclusively excitatory. Post-AAV-injection behavioural testing was performed within 1–5 weeks for experiments using AAV. flex.hM3Dq-mCherry, and within 4–6 days for experiments using AAV.flex.TeLC.eGFP. For anatomical analyses of inhibitory synaptic input on to GRPR-INs AAV-Brainbow2 injections were performed unilaterally into the L3 and L5 segments and mice were perfused 2–3 weeks post-surgery. For electrophysiological studies, AAV.flex.ChR2-eYFP, on some occasions combined with AAV.FRT.mCherry, was injected unilaterally or bilaterally into the L3 and/or L5 segments and spinal cord slices were prepared from the mice 1–3 weeks post-surgery.

## Intraplantar CFA injections

Mice were briefly anaesthetised with 1–2% isoflurane, the plantar surface of the hindpaw ipsilateral to the spinal AAV injection was wiped with 70% ethanol and 20 µl of 1 mg/ml CFA was injected subcutaneously. Behavioural testing was performed prior to (pre-CFA baseline) and 2 days following CFA injections.

## SNI surgery

Mice were anaesthetised with 1–2% isoflurane, an incision was made in the skin over the thigh ipsilateral to the spinal AAV injection and the underlying muscle was blunt dissected to reveal the sciatic nerve. The tibial and common peroneal branches were identified, and 7–0 Mersilk (Ethicon, Puerto Rico) was used to apply two tight ligatures 2–3 mm apart on each nerve branch. The length of nerve between the ligatures was then removed and the wound was closed. Great care was taken to avoid damage to the sural branch of the sciatic nerve during the surgery. Behavioural testing was performed prior to (pre-SNI baseline) and from 2 to 4 weeks following SNI surgery.

## Drug administration

CNO (Tocris Bioscience, Bristol, UK) dissolved in a 10% dimethyl sulphoxide (DMSO)/90% sterile saline mixture was injected intraperitoneally (i.p.) at a dose of 5 mg/kg; 10% DMSO/90% sterile saline mixture alone was used as a vehicle control. In some cases, we used CNO-dihydrochloride (Tocris Bioscience), dissolved in 10% water/90% sterile saline at a dose of 5 mg/kg, with 10% water/90% sterile saline as a vehicle control. For some experiments, the Y1 antagonist BMS 193885 (Bio-Techne, Abingdon, UK) dissolved in a 40% PEG-400/60% sterile saline mixture was co-injected i.p. at a dose of 10 mg/kg with CNO; co-injection of the respective vehicles for CNO and BMS 193885 was used as a control. Gabapentin (Sigma-Aldrich) was injected i.p. at a dose of 100 mg/kg. The timing of CNO or gabapentin injections for CPP testing are described in the relevant section below. For all other behavioural testing, CNO, CNO + BMS 193885 or vehicle were injected a minimum of 30 min prior to the start of testing, and all testing was completed within a maximum of 5 hr following injection.

## Behavioural testing

### von Frey test (noxious punctate mechanical sensitivity)

Mice were placed in a plastic enclosure with mesh flooring and allowed to acclimatise for at least 45 min. von Frey filaments of logarithmically incremental stiffness (range 0.01–4 g) were applied to the plantar surface of the hindpaw and the 50% MWT was determined using Dixon's up-down method (*Chaplan et al., 1994*; *Dixon, 1980*). Briefly, filaments were applied sequentially, beginning with the mid-range filament (0.4 g), and the presence or absence of a withdrawal response (lifting and/or shaking of the paw) was noted. If a withdrawal response was observed, the next lowest filament was used subsequently; if no response was observed, the next highest filament was used subsequently. Testing continued until a series of six filaments had been applied from the point when the response threshold was first crossed, and the 50% MWT was calculated using the formula 50% MWT $= (10^{[Xf+k\delta]})/10,000$, where $Xf$ = log value of the final filament applied, $k$ = tabular value (taken from *Chaplan et al., 1994*) based on the pattern of six positive/negative responses and $\delta$ = mean difference (in log units) of the range of filaments used (0.323). When testing mice that had undergone SNI, care was taken to apply von Frey filaments to the sural territory of the hindpaw.

### Hargreaves test (noxious heat sensitivity)

Mice were placed in plastic enclosures on a raised glass platform warmed to 25°C and allowed to acclimatise for at least 30 min. A radiant heat source (IITC, CA, USA) set to 25% active intensity was

targeted to the plantar surface of the hindpaw to be tested (using an angled mirror and guide light), and the time until paw withdrawal from the heat source (withdrawal latency) was noted. Testing of ipsi- and contralateral paws was alternated with at least 3-min interval between consecutive tests, and a cut-off time of 25 s was used to prevent tissue damage. Each hindpaw was tested five times, and the average withdrawal latency calculated. When testing mice that had undergone SNI, care was taken to target the heat source to the sural territory of the hindpaw.

### Dry ice test (noxious cold sensitivity)

Mice were placed in plastic enclosures on a raised 5-mm-thick glass platform at room temperature and allowed to acclimatise for at least 45 min. A dry ice pellet of ~1 cm diameter was applied to the underside of the glass directly below the hindpaw to be tested, and the withdrawal latency was recorded (*Brenner et al., 2012*). Care was taken to ensure that the plantar surface of the hindpaw was in direct contact with the glass prior to testing. Testing of ipsi- and contralateral paws was alternated with at least 3-min interval between consecutive tests, and a cut-off time of 25 s was used to prevent tissue damage. Each hindpaw was tested five times, and the average withdrawal latency calculated.

### Acetone evaporation test (noxious cold sensitivity)

Mice were placed in plastic enclosure with mesh flooring and allowed to acclimatise for at least 30 min. A 10 µl droplet of acetone was applied to the sural territory of the plantar hindpaw and the total amount of time spent shaking, lifting, and licking the paw within 30 s of acetone application was recorded using a stopwatch. Each hindpaw was tested three times, with at least 3-min interval between consecutive tests, and the average total response time calculated.

### Rotarod test (motor co-ordination)

Mice were placed into the rotarod apparatus (IITC, CA, USA), which was programmed to accelerate from 4 to 40 rpm over 5 min. Mice were allowed two trial runs prior to performing four test runs, and the average maximum rpm attained was calculated from the test runs.

### Pruritogen-evoked itch test

Mice were acclimatised for 30 min in plastic enclosures surrounded by angled mirrors to provide unobstructed views of the targeted hindlimb, and were then video recorded for 30 min (pre-CQ). They then received 10 µl of 1% CQ dissolved in phosphate-buffered saline (PBS) via intradermal injection into the calf ipsilateral to the spinal AAV injection (which had been shaved 24 hr prior to testing). Successful intradermal injection of CQ was assessed by the appearance of a skin bleb at the injection site. Mice were video recorded for 30 min following CQ injection (post-CQ). The amount of time spent biting the injected area was scored offline either manually with a stopwatch or using BORIS event logging software (freely available, https://www.boris.unito.it/; *Friard et al., 2016*). Videos were viewed at one-quarter speed for analysis.

### CPP test

To test for ongoing neuropathic pain, a 3-day conditioning protocol using a biased chamber assignment was used for CPP testing as described previously (*Cooper et al., 2022*). The custom 3-chamber CPP apparatus consisted of two conditioning side chambers (170 × 150 mm) connected by a centre chamber (70 × 75 mm), 180 mm tall, with infrared-transparent plastic lids (QD Plastics, Dumbarton, UK). Mice were able to discriminate between chambers using visual (vertical vs. horizontal black-and-white striped walls) and sensory (rough vs. smooth textured floor) cues. On day 1 (acclimation, 7 days after SNI surgery), mice had free access to explore all chambers for 30 min. On days 2 and 3 (preconditioning), mice were again allowed to freely explore for 30 min whilst their position was recorded using an infrared camera and AnyMaze 7.16 software (Stoelting, USA). To avoid pre-existing chamber bias, mice spending more than 90% or less than 5% of time in either side chamber during preconditioning were excluded (1 mouse from each experimental group). For conditioning (days 4–6), each morning, mice received i.p. vehicle injection, were returned to their home cage for 5 min, then confined to their preferred side chamber for 30 min. Four hours later, mice received i.p. CNO (5 mg/kg) or gabapentin (100 mg/kg), were returned to their home cage for 5 min, and then placed in their

non-preferred chamber for 30 min. On test day (day 7), mice could freely explore all chambers whilst their position was recorded, as during pre-conditioning, for 30 min. Difference scores were calculated as the time spent in each chamber on test day minus the mean time spent during pre-conditioning. We have previously shown that CNO given at a much lower dose (0.2 mg/kg) to mice in which spinal GRPR neurons expressed hM3Dq resulted in itch- and pain-related behaviours that started within 5 min of administration (*Polgár et al., 2023*). It is therefore very likely that NPY neurons would have been activated throughout the conditioning period for CNO. In addition, the timecourse of action of CNO and gabapentin are likely to be similar, since both were administered i.p., and there was a clear preference for the chamber paired with gabapentin.

## Noxious heat and pruritic induction of Fos

Mice were injected i.p. with CNO or vehicle 30 min prior to receiving noxious heat or pruritic stimulation under brief isoflurane anaesthesia, ipsilateral to spinal AAV.flex.hM3Dq-mCherry injection. The noxious heat stimulus was immersion of the hindpaw into 52°C water for 15 s. The pruritic stimulus was intradermal injection of CQ into the calf as described above, following shaving of the leg 24 hr prior to CQ injection. These mice were fitted with Elizabethan collars to prevent Fos induction through scratching and/or biting of the injected area. Mice were transcardially perfused under deep terminal general anaesthesia 2 hr after stimulation, and spinal cord tissue was processed for imaging and analysis as described below.

## Electrophysiology

Electrophysiological studies were performed on spinal cord slices from Npy-Cre or Npy-Cre;$Grpr^{FLPo}$ mice that had received an intraspinal injection of AAV.flex.ChR2-eYFP or AAV.flex.ChR2-eYFP combined with AAV.FRT.mCherry, respectively, 1–3 weeks prior to recordings. Additional recordings were performed on spinal cord slices from $Grpr^{CreERT2}$;Ai9 mice. Recordings were made from 51 cells (39 from female, 12 from male mice) from spinal cord slices obtained from 21 Npy-Cre mice (16 female, 5 male) and an additional 11 cells from 5 female Npy-Cre;$Grpr^{FLPo}$ mice and 10 cells from 5 female $Grpr^{CreERT2}$;Ai9 mice, that were aged 5–11 weeks. Spinal cord slices were prepared as described previously (*Dickie et al., 2019*). Mice were decapitated under isoflurane anaesthesia and the spinal cord removed in ice-cold dissection solution. In some cases decapitation was performed following transcardial perfusion with ice-cold dissection solution under terminal anaesthesia with pentobarbital (20 mg i.p.). The lumbar region was embedded in an agarose block and 300 μm parasagittal or 350 μm transverse slices were cut with a vibrating blade microtome (Thermo Scientific Microm HM 650V, Loughborough, UK; Leica VT1200s, Milton Keynes, UK; or Campden Instruments 7000smz-2, Loughborough, UK). Slices were then allowed to recover for at least 30 min in recording solution at room temperature. In some cases, slices were placed in an *N*-methyl-D-glucamine (NMDG)-based recovery solution at ~32°C for 15 min (*Ting et al., 2014*) before being placed in a modified recording solution at room temperature for at least 30 min. The solutions used contained the following (in mM); dissection, 251.6 sucrose, 3.0 KCl, 1.2 $NaH_2PO_4$, 0.5 $CaCl_2$, 7.0 $MgCl_2$, 26.0 $NaHCO_3$, 15.0 glucose; NMDG recovery, 93.0 NMDG, 2.5 KCl, 1.2 $NaH_2PO_4$, 0.5 $CaCl_2$, 10.0 $MgSO_4$, 30.0 $NaHCO_3$, 25.0 glucose, 5.0 Na-ascorbate, 2.0 thiourea, 3.0 Na-pyruvate, and 20.0 HEPES (4-(2-hydroxyethyl)-1-piperazine ethanesulfonic acid); modified recording, 92.0 NaCl, 2.5 KCl, 1.2 $NaH_2PO_4$, 2.0 $CaCl_2$, 2.0 $MgSO_4$, 30.0 $NaHCO_3$, 25.0 glucose, 5.0 Na-ascorbate, 2.0 thiourea, 3.0 Na-pyruvate, and 20.0 HEPES; and recording, 125.8 NaCl, 3.0 KCl, 1.2 $NaH_2PO_4$, 2.4 $CaCl_2$, 1.3 $MgCl_2$, 26.0 $NaHCO_3$, and 15.0 glucose. All solutions were bubbled with 95% $O_2$/5% $CO_2$.

Whole-cell patch-clamp recordings were made from ChR2-eYFP+ or ChR2-eYFP− neurons in Npy-Cre tissue, from mCherry+ cells in Npy-Cre;$Grpr^{FLPo}$ tissue, or from tdTom+ cells in $Grpr^{CreERT2}$;Ai9 tissue in the SDH, using patch pipettes that had a typical resistance of 3–7 MΩ. Data were recorded and acquired with a Multiclamp 700B amplifier and pClamp 10 software (both Molecular Devices, Wokingham, UK), and were filtered at 4 kHz and digitised at 10 kHz.

To validate the optogenetic activation of NPY neurons, recordings were made from ChR2-eYFP + cells using a K-based intracellular solution. The presence of optogenetically activated currents were observed in voltage clamp mode, from a holding potential of −70 mV, by illuminating the slice with brief (1–4 ms) pulses of blue light, generated by a 470-nm LED (CoolLED pE-100, Andover, UK) and delivered via the microscope objective. The ability to optogenetically drive action potential firing in

NPY neurons was similarly tested by applying brief pulses of blue light while the membrane potential was held around −60 mV in current clamp mode.

Inhibition from NPY cells to other cells in the SDH was investigated in tissue from Npy-Cre mice, by recording from ChR2-eYFP-negative cells that were within the region of the viral injection, as determined by eYFP expression, and applying pulses of blue light as detailed above. Inhibition of GRPR cells by NPY cells was similarly assessed by recording from mCherry+ GRPR cells in slices from Npy-Cre;$Grpr^{FLPo}$ mice. Cells were classed as receiving optogenetically evoked IPSCs (oIPSCs) if there was a clear reliably evoked inward current (CsCl-based intracellular, $V_{hold}$ −70 mV), or outward current (K-based intracellular, $V_{hold}$ −40 mV; Cs-methanesulfonate-based intracellular, $V_{hold}$ 0 mV), that was time locked to the light pulse. In a subset of recordings, using CsCl-based intracellular solution ($V_{hold}$ −70 mV), the optogenetically evoked postsynaptic currents were confirmed to be non-glutamatergic by bath application of NBQX (10 μM) and D-APV (30 μM). The nature of the oIPSCs recorded in unlabelled (CsCl-based intracellular, $V_{hold}$ −70 mV) and GRPR (Cs-methanesulfonate-based intracellular, $V_{hold}$ 0 mV) cells was investigated by bath application of the GABA antagonist, gabazine (300 nM) and the glycine antagonist, strychnine (300 nM); in the case of unlabelled cells this was done in the presence of NBQX and D-APV. oIPSCs were evoked six times (0.05 Hz) (Baseline) prior to the application of gabazine or strychnine, which was added to the recording solution and washed into the recording chamber for 5 min before and during the recording of a further six oIPSCs. Strychnine was then added to gabazine or gabazine added to strychnine and following a further 5 min wash in period six oIPSCs were recorded. oIPSCs were classified as 'sensitive' to gabazine and/or strychnine if the drug reduced the mean peak amplitude of the oIPSC to a level that was less than 2 SD of the mean peak amplitude recorded during baseline or the previous drug application, and 'insensitive' if this threshold was not met. The monosynaptic nature of the oIPSCs was tested in a subset of GRPR cells, by investigating whether oIPSCs that were abolished by the application of TTX (0.5 μM) could be rescued by the addition of 4-aminopyridine (4-AP; 100 μM; *Petreanu et al., 2009*). To test whether the release of NPY from NPY cells may contribute to the inhibition of GRPR cells, patch clamp recordings were made from tdTom+ GRPR cells from $Grpr^{CreERT2}$;Ai9 mice to assess responses to the NPY Y$_1$ receptor agonist, [Leu$^{31}$,Pro$^{34}$]-neuropeptide Y (300 nM), which was bath applied in the presence of TTX (0.5 μM), bicuculline (10 μM), and strychnine (1 μM) (K-based intracellular, $V_{hold}$ −50 mV). Following a 1-min baseline period, [Leu$^{31}$,Pro$^{34}$]-neuropeptide Y was applied for 4 min, and cells where classified as responders if [Leu$^{31}$,Pro$^{34}$]-neuropeptide Y resulted in an outward current of 5 pA or greater and were classified as non-responders if this threshold was not reached.

The intracellular solutions used contained the following (in mM): K-based, 130.0 K-gluconate, 10.0 KCl, 2.0 MgCl$_2$, 10.0 HEPES , 0.5 EGTA (ethylene glycol-bis(β-aminoethyl ether)-N,N,N',N'-tetraacetic acid), 2.0 ATP-Na$_2$, 0.5 GTP-Na, and 0.2% Neurobiotin, pH adjusted to 7.3 with 1.0 M KOH; CsCl-based, 130.0 CsCl, 1.0 MgCl$_2$, 10.0 HEPES, 10.0 EGTA, 5.0 *N*-(2,6-dimethylphenylcarbamoylmethyl) triethylammonium bromide (QX-314-Br), 2.0 ATP-Na2, 0.3 GTP-Na, and 0.2% Neurobiotin, pH adjusted to 7.35 with 1.0 M CsOH; Cs-methylsulfonate-based, 120.0 Cs-methylsulfonate, 10.0 Na-methylsulfonate, 10.0 EGTA, 1.0 CaCl$_2$, 10.0 HEPES, 5.0 *N*-(2,6-dimethylphenylcarbamoylmethyl) triethylammonium chloride (QX-314-Cl), 2.0 Mg$_2$-ATP, and 0.2% Neurobiotin, pH adjusted to 7.2 with 1.0 M CsOH.

All chemicals were obtained from Sigma except: TTX, QX-314-Br, QX-314-Cl (Alomone, Jerusalem, Israel), Gabazine, NBQX (Abcam, Cambridge, UK), sucrose, glucose, NaH$_2$PO$_4$ (VWR, Lutterworth, UK), D-APV, [Leu$^{31}$,Pro$^{34}$]-neuropeptide Y (Tocris, Abingdon, UK), and Neurobiotin (Vector Labs, Peterborough, UK).

## Immunohistochemistry

Animals were terminally anaesthetised with pentobarbital (20 mg i.p.) and transcardially perfused with 4% freshly depolymerised formaldehyde. The spinal cord was then dissected out and post-fixed in the same fixative for 2 hr, and 60-μm-thick transverse or sagittal sections from appropriate lumbar sections were cut on a vibrating blade microtome. Sections were immersed in 50% ethanol for 30 min to enhance antibody penetration before incubation in appropriate primary and secondary antibodies at 4°C for 72 and 24 hr, respectively. Details of the antibodies used in this study can be found in the Key Resources Table. Sections were continuously agitated during antibody incubation and washed three times in PBS that contained 0.3 M NaCl following each incubation. Following the final PBS wash, sections were mounted on slides in Vectashield anti-fade mounting medium (Vector Laboratories, CA,

USA). In some cases, lumbar dorsal root ganglia (DRGs) were also removed and processed intact for immunohistochemistry (IHC) before whole-mounting on slides.

## Fluorescent in situ hybridisation

Multiple-labelling FISH was performed using RNAscope probes and RNAscope fluorescent multiplex reagent kit 320850 (ACD BioTechne, CA, USA). Mice were deeply anaesthetised, the spinal cord was rapidly removed by hydraulic extrusion and the lumbar enlargement was excised and snap-frozen on dry ice. Lumbar segments were then embedded in OCT mounting medium and 12-μm-thick transverse sections were cut on a Leica CM 1950 cryostat (Leica, Milton Keynes, UK). Sections were mounted non-sequentially (such that sections on the same slide were at least four apart) onto SuperFrost Plus slides (VWR, Lutterworth, UK) and air-dried. The slides were then reacted according to the manufacturer's recommended protocol, using probes against *Cre* and *Npy* that were revealed with Atto 550 and Alexa 647, respectively. Further details of the probes used can be found in the Key Resources Table. Sections were mounted using Prolong-Glass anti-fade medium containing NucBlue (Hoescht 33342) nuclear stain (Thermo Fisher Scientific, Paisley, UK).

## Image acquisition and analysis

IHC and FISH slides were imaged with a Zeiss LSM 710 confocal microscope system equipped with Ar multi-line, 405 nm diode, 561 nm solid-state and 633 nm HeNe lasers. For analyses of AAV.flex. FP, AAV.flex.hM3Dq-mCherry, or AAV.flex.TeLC.eGFP-labelled cells image stacks were taken through a ×40 oil-immersion objective (NA = 1.3) at a z-separation of 1 μm. For assessment of spinal injection sites and analysis of mCherry expression in DRGs of AAV.flex.hM3Dq-mCherry-injected mice, image stacks were taken through a ×10 objective (NA = 0.3) at a z-separation of 2 μm. Image analysis was performed using Neurolucida software (MBF Bioscience, VT, USA).

For comparison of tdTomato or eGFP expression with NPY immunoreactivity in AAV.flex.FP-injected Npy-Cre mice, and in AAV.flex.eGFP-injected Npy-Cre;Ai9 mice, a modified optical disector method was used (*Polgár et al., 2004*). All structures labelled with the neuronal marker NeuN that had their bottom surface between reference and look-up sections separated by 10 μm (10 optical sections) were marked within the injection site in laminae I–III. The NPY-IR, eGFP, and/or tdTomato channels were then viewed separately and sequentially, and the NeuN profiles were marked as positive or negative as appropriate. As NPY-IR varies greatly from cell to cell and is often observed as discrete clumps within the perikaryal cytoplasm (*Boyle et al., 2017*; *Iwagaki et al., 2016*), cells were classified as NPY-positive if clear above-background signal was observed in at least three consecutive optical sections. The same optical disector method was used to identify NeuN-positive cells, and compare mCherry, Fos and Pax2 or mCherry, Fos and NPY immunoreactivity in these cells in vehicle- or CNO-dosed AAV.flex.hM3Dq-mCherry spinal-injected Npy-Cre mice, as well as eGFP and NPY immunoreactivity in the AAV.flex.TeLC.eGFP spinal-injected Npy-Cre mice. A similar method was used to compare mCherry, Fos and Pax2 in NeuN-positive cells in the vehicle- or CNO-dosed AAV.flex. hM3Dq-mCherry spinal-injected Npy-Cre mice that had received noxious heat or pruritic stimuli 2 hr prior to perfusion; however, in this case the reference and look-up sections were separated by 20 μm (20 optical sections) and analyses were restricted to the somatotopically relevant areas of laminae I and II (medial half for noxious heat stimulation of hindpaw, middle third for CQ injection into calf). For all of these analyses, markers from previously assessed channels were hidden as the channels were analysed in sequence, to prevent bias.

For comparison of tdTomato and eGFP or mCherry expression with neurochemical markers of inhibitory interneuron populations in AAV.flex.eGFP-injected Npy-Cre;Ai9 or AAV.flex.hM3Dq-mCherry-injected Npy-Cre mice, respectively, all cells expressing the neurochemical markers and FP(s) in laminae I and II were marked throughout the whole section thickness. Each channel was again assessed separately and sequentially, with markers from previously assessed channels hidden to prevent bias. Only cells with the maximal profile of their soma contained within the z stack were counted. mCherry-positive cells in whole-mounted DRGs from AAV.flex.hM3Dq-mCherry-injected Npy-Cre mice that underwent SNI surgery were counted using the same method.

For analysis of inhibitory synapses onto GRPR-INs, nine mTFP-labelled cells in total were selected from tissue of three AAV-Brainbow2-injected *Grpr^CreERT2^* mice immunostained for mTFP, VGAT, gephyrin and NPY or Dynorphin B (3 per animal for Dynorphin B and 2, 3, and 4 cells for NPY analyses).

Cell selection was performed before the staining for axonal markers was visualised and was based on the completeness of dendritic labelling and separation from other nearby mTFP-labelled neurons. The selected cells were scanned through a ×63 oil-immersion lens (numerical aperture 1.4) at a z-separation of 0.3 μm. Z-series were obtained from as much of the dendritic tree as was visible in the section. For analysis, the mTFP and gephyrin channels were initially viewed and the cell bodies and dendritic trees were traced based on the mTFP signal. The locations of all gephyrin puncta associated with the cell body and dendritic tree were then plotted. The VGAT channel was then viewed and the presence or absence of an apposed VGAT-IR bouton was noted for each gephyrin punctum. Finally, the remaining channel (corresponding to NPY or dynorphin B) was revealed and the presence or absence of peptide staining was noted for each of the VGAT boutons that contacted a gephyrin punctum on the selected cell. To determine the frequency of all boutons arising from inhibitory interneurons that were positive for each of the neuropeptides, we sampled from those VGAT-IR boutons in the vicinity of the mTFP-labelled cell. A 4 × 4 μm grid was applied within a box drawn to include the entire dendritic tree of the cell. Only the VGAT channel was viewed initially and in each successive grid square, the VGAT-IR bouton nearest the bottom right of the square was selected. The presence or absence of NPY or dynorphin B was then recorded for each of these selected VGAT-IR boutons.

For comparison of *Cre* and *Npy* mRNA in FISH sections, all NucBlue profiles within a single optical section were marked throughout laminae I–III. The *Cre* and *Npy* channels were then viewed separately and sequentially and cells were marked as positive if they contained ≥4 labelled transcript particles. Markers from the mRNA channel that was assessed first were hidden during assessment of the second channel to prevent bias.

## Quantification and statistical analysis

Data are reported as mean ± standard error of the mean, unless stated otherwise. Statistical analyses were performed using Prism software (v7, v8, and v9, GraphPad Software, CA, USA). Behavioural and anatomical analyses involving drug treatments were analysed blind to treatment group. Behavioural analyses involving AAV.flex.TeLC.eGFP-mediated silencing and AAV.flex.eGFP-injected controls were analysed blind to the AAV injected. The statistical tests used for each experiment, including tests for multiple comparisons, are given in the appropriate figure legends. A p value of <0.05 was considered significant, and significance markers are denoted within figures as follows: *p < 0.05, **p < 0.01, ***p < 0.001, ****p < 0.0001.

## Acknowledgements

This research was funded in whole, or in part, by the Wellcome Trust (Grant numbers 102645/Z/13/Z and 219433/Z/19/Z), the Biotechnology and Biological Sciences Research Council (Grant numbers BB/N006119/1 and BB/P007996/1) and the Medical Research Council (Grant numbers MR/S002987/1, MR/T01072X/1, and MR/V033638/1). For the purpose of Open Access, the authors have applied a CC BY public copyright licence to any Author Accepted Manuscript version arising from this submission. We are grateful to R Kerr, C Watt, and I Plenderleith for expert technical assistance, to Yan-Gang Sun for the gift of *Grpr*^CreERT2 and *Grpr*^FLPo mice, to Dr Philippe Ciofi for the dynorphin B antibody and to Dr Mark Hoon for helpful discussion.

## Additional information

### Funding

| Funder | Grant reference number | Author |
| --- | --- | --- |
| Wellcome Trust | 102645/Z/13/Z | Andrew J Todd |
| Wellcome Trust | 219433/Z/19/Z | Andrew J Todd |
| Biotechnology and Biological Sciences Research Council | BB/N006119/1 | John S Riddell Andrew J Todd |

| Funder | Grant reference number | Author |
|---|---|---|
| Biotechnology and Biological Sciences Research Council | BB/P007996/1 | David I Hughes John S Riddell Andrew J Todd |
| Medical Research Council | MR/S002987/1 | John S Riddell Andrew J Todd |
| Medical Research Council | MR/T01072X/1 | Gregory A Weir |
| Medical Research Council | MR/V033638/1 | John S Riddell Andrew J Todd |

The funders had no role in study design, data collection, and interpretation, or the decision to submit the work for publication. For the purpose of Open Access, the authors have applied a CC BY public copyright license to any Author Accepted Manuscript version arising from this submission.

## Author contributions

Kieran A Boyle, Conceptualization, Data curation, Formal analysis, Investigation, Writing – original draft, Writing - review and editing; Erika Polgar, Maria Gutierrez-Mecinas, Allen C Dickie, Andrew H Cooper, Andrew M Bell, Conceptualization, Formal analysis, Investigation, Writing – original draft; Evelline Jumolea, Adrian Casas-Benito, Formal analysis, Writing – original draft; Masahiko Watanabe, Resources, Writing – original draft; David I Hughes, Investigation, Writing – original draft; Gregory A Weir, Conceptualization, Writing – original draft; John S Riddell, Conceptualization, Funding acquisition, Writing – original draft; Andrew J Todd, Conceptualization, Formal analysis, Supervision, Funding acquisition, Investigation, Writing – original draft, Project administration, Writing - review and editing

## Author ORCIDs

Kieran A Boyle (ID) http://orcid.org/0000-0003-3201-6149
Allen C Dickie (ID) http://orcid.org/0000-0002-6339-2801
Andrew H Cooper (ID) http://orcid.org/0000-0003-4737-9364
Andrew M Bell (ID) http://orcid.org/0000-0001-6510-4423
Adrian Casas-Benito (ID) http://orcid.org/0000-0002-4421-5551
Masahiko Watanabe (ID) http://orcid.org/0000-0001-5037-7138
David I Hughes (ID) http://orcid.org/0000-0003-1260-3362
Andrew J Todd (ID) http://orcid.org/0000-0002-3007-6749

## Ethics

All experiments were approved by the Ethical Review Process Applications Panel of the University of Glasgow, and were carried out in accordance with the European Community directive 86/609/EC, the UK Animals (Scientific Procedures) Act 1986 and ARRIVE guidelines.

Reviewer #1 (Public Review): https://doi.org/10.7554/eLife.86633.3.sa1
Reviewer #2 (Public Review): https://doi.org/10.7554/eLife.86633.3.sa2
Reviewer #3 (Public Review): https://doi.org/10.7554/eLife.86633.3.sa3
Author Response https://doi.org/10.7554/eLife.86633.3.sa4

---

# Additional files

## Supplementary files
• MDAR checklist

## Data availability

Data can be accessed from an open repository at the following link: http://dx.doi.org/10.5525/gla.researchdata.1448. This study did not generate any new materials, reagents, or code.

The following dataset was generated:

| Author(s) | Year | Dataset title | Dataset URL | Database and Identifier |
|---|---|---|---|---|
| Boyle KA, Polgár E, Gutierrez-Mecinas M, Dickie AC, Cooper AH, Bell AM, Jumolea ME, Casas-Benito A, Watanabe M, Hughes DI, Weir GA, Riddell JS, Todd AJ | 2023 | Neuropeptide Y-expressing dorsal horn inhibitory interneurons gate spinal pain and itch signalling | http://dx.doi.org/10.5525/gla.researchdata.1448 | Enlighten Research Data, 10.5525/gla.researchdata.1448 |

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

# Appendix 1

## Appendix 1—key resources table

| Reagent type (species) or resource | Designation | Source or reference | Identifiers | Additional information |
|---|---|---|---|---|
| Strain, strain background (*Mus musculus*) | STOCK Tg(Npy-cre) RH26Gsat/Mmucd (Npy-Cre) | GENSAT/MMRRC | Cat#: 34810-UCD RRID: MMRRC_34810-UCD | |
| Strain, strain background (*Mus musculus*) | B6.Cg-Gt(ROSA)26Sor<sup>tm9(CAG-tdTomato)Hze</sup>/J (Ai9) | Prof. Hongkui Zheng | Cat#: 007909 RRID: IMSR_JAX:007909 | Available from The Jackson Laboratory, ME, USA |
| Strain, strain background (*Mus musculus*) | GRPR-iCreERT2 (*Grpr<sup>CreERT2</sup>*) | Dr. Yan-Gang Sun | | *Liu et al., 2019* |
| Strain, strain background (*Mus musculus*) | Grpr<sup>FLPo</sup> | Dr. Yan-Gang Sun | | *Mu et al., 2017* |
| Strain, strain background (*Mus musculus*) | B6;129S-Pdyn<sup>tm1.1(cre)Mjkr</sup>/LowlJ (*Pdyn<sup>Cre</sup>*) | The Jackson Laboratory, ME, USA | Cat#: 027958 RRID: IMSR_JAX:027958 | |
| Strain, strain background (*Adeno-associated virus*) | AAV1 pCAG-FLEX-EGFP-WPRE (AAV.flex.eGFP; 1.72 × 10⁹ GC) | Penn Vector Core, PA, USA; available from Addgene, MA, USA (deposited by Hongkui Zeng) | Cat#: 51502 RRID: Addgene_51502 | *Oh et al., 2014* |
| Strain, strain background (*Adeno-associated virus*) | AAV1 pCAG-FLEX-tdTomato-WPRE (AAV.flex.tdTomato; 1.76 × 10⁹ GC) | Penn Vector Core, PA, USA; available from Addgene, MA, USA (deposited by Hongkui Zeng) | Cat#: 51503 RRID: Addgene_51503 | *Oh et al., 2014* |
| Strain, strain background (*Adeno-associated virus*) | pAAV1-EF1a-double floxed-hChR2(H134R)-EYFP-WPRE-HGHpa (AAV.ChR2-eYFP; 5.09 × 10⁸ GC) | Penn Vector Core, PA, USA; available from Addgene, MA, USA (deposited by Karl Deisseroth) | Cat#: 20298 RRID: Addgene_20298 | |
| Strain, strain background (*Adeno-associated virus*) | rAAV2/hSyn-DIO-hm3D-mcherry (AAV.flex.hM3Dq-mCherry; 3.8 × 10⁸ GC) | University of North Carolina Vector Core, NC, USA; available from Addgene, MA, USA (deposited by Bryan Roth) | Cat#: 44361 RRID: Addgene_44361 | *Krashes et al., 2011* |
| Strain, strain background (*Adeno-associated virus*) | ssAAV-2/2-hSyn1-dlox-hM3D(Gq)_mCherry(rev)-dlox-WPRE-hGHp(A) (AAV.flex.hM3Dq-mCherry; 7.65 × 10⁸ GC) | Viral Vector Facility, University of Zurich, Switzerland | Cat#: v89-2 | |
| Strain, strain background (*Adeno-associated virus*) | ssAAV-2/2-hSyn1-dlox-EGFP(rev)-dlox-WPRE-hGHp(A) (AAV.flex.eGFP; 2 × 10⁸ GC) | Viral Vector Facility, University of Zurich, Switzerland | Cat#: v115-2 | |
| Strain, strain background (*Adeno-associated virus*) | pssAAV-2-hSyn1-chI-dlox-EGFP_2A_FLAG_TeTxLC(rev)-dlox-WPRE-SV40p(A) (AAV.flex.TeLC.eGFP; 2 × 10⁸ GC) | Viral Vector Facility, University of Zurich, Switzerland | Cat#: v322-2 | |
| Strain, strain background (*Adeno-associated virus*) | ssAAV-8/2-hSyn1-dFRT-mCherry(rev)-dFRT-WPRE-hGHp(A) (AAV.FRT.mCherry; 8.7 × 10⁸ GC) | Viral Vector Facility, University of Zurich, Switzerland | Cat#: v188-8 | |

*Appendix 1 Continued on next page*

*Appendix 1 Continued*

| Reagent type (species) or resource | Designation | Source or reference | Identifiers | Additional information |
|---|---|---|---|---|
| Strain, strain background (*Adeno-associated virus*) | AAV9-EF1a-BbChT (AAV-Brainbow2; 1.5–5.96 × 10$^7$ GC) | Addgene, MA, USA; Deposited by Dawen Cai & Joshua Sanes | Cat#: 45186-AAV9 RRID: Addgene_45186 | *Cai et al., 2013* |
| Antibody | anti-Calretinin (Goat polyclonal) | SWANT, Bellinoza, Switzerland | Cat#: CG1 RRID: AB_10000342 | IF (1:1000) |
| Antibody | anti-cFOS (Goat polyclonal) | Santa Cruz Biotechnology Inc, CA, USA | Cat#: sc-52-G RRID: AB_2629503 | IF (1:2000) |
| Antibody | anti-DynorphinB (Rabbit polyclonal) | Dr. Philippe Ciofi, INSERM, France | Cat#: IS-35 RRID: AB_2819033 | IF (1:500) |
| Antibody | anti-Galanin (Rabbit polyclonal) | Peninsula Laboratories, CA, USA | Cat#: T-4334 RRID: AB_518348 | IF (1:1000) |
| Antibody | anti-Gephyrin (Mouse monoclonal) | SynapticSystems, Göttingen, Germany | Cat#: 147021 RRID: AB_2232546 | IF (1:500) |
| Antibody | anti-GFP (Chicken polyclonal) | Abcam plc., Cambridge, UK | Cat#: ab13970 RRID: AB_300798 | IF (1:1000) |
| Antibody | anti-mCherry (Chicken polyclonal) | Abcam plc., Cambridge, UK | Cat#: ab205402 RRID: AB_2722769 | IF (1:10,000) |
| Antibody | anti-mCherry (Rat monoclonal) | Invitrogen; Thermo Fisher Scientific, UK | Cat#: M11217 RRID: AB_2536611 | IF (1:1000) |
| Antibody | anti-mTFP (Rat polyclonal) | Kerafast Inc, Boston, MA, USA | Cat#: EMU108 | IF (1:500) |
| Antibody | anti-NeuN (Chicken polyclonal) | SynapticSystems, Göttingen, Germany | Cat#: 266006 RRID: AB_2571734 | IF (1:1000) |
| Antibody | anti-NeuN (Guinea pig polyclonal) | SynapticSystems, Göttingen, Germany | Cat#: 266004 RRID: AB_2619988 | IF (1:500) |
| Antibody | anti-nNOS (Rabbit polyclonal) | MilliporeSigma, MA, USA | Cat#: 07-571 RRID: AB_310722 | IF (1:2000) |
| Antibody | anti-NPY (Rabbit polyclonal) | Peninsula Laboratories, CA, USA | Cat#: T-4070 RRID: AB_518504 | IF (1:1000) |
| Antibody | anti-Parvalbumin (Guinea pig polyclonal) | Frontier Institute Co. Ltd, Hokkaido, Japan | Cat#: PV-GP-Af1000 RRID: AB_2336938 | IF (1:2500) |
| Antibody | anti-PAX2 (Rabbit polyclonal) | Invitrogen; Thermo Fisher Scientific, UK | Cat#: 71-6000 RRID: AB_2533990 | IF (1:1000) |
| Antibody | anti-PAX2 (Rabbit polyclonal) | MilliporeSigma, MA, USA | Cat#: HPA047704 RRID: AB_2636861 | IF (1:200) |
| Antibody | anti-VGAT (Goat polyclonal) | Frontier Institute Co. Ltd, Hokkaido, Japan | Cat#: VGAT-Go-Af620 RRID: AB_2571623 | IF (1:1000) |
| Antibody | Alexa Fluor 488 Donkey Anti-Chicken IgY (Donkey polyclonal) | Jackson ImmunoResearch, PA, USA | Cat#: 703-545-155 RRID: AB_2340375 | IF (1:500) |
| Antibody | Alexa Fluor 488 Donkey Anti-Goat IgG (Donkey polyclonal) | Jackson ImmunoResearch, PA, USA | Cat#: 705-545-003 RRID: AB_2340428 | IF (1:500) |
| Antibody | Alexa Fluor 488 Donkey Anti-Rabbit IgG (Donkey polyclonal) | Jackson ImmunoResearch, PA, USA | Cat#: 711-545-152 RRID: AB_2313584 | IF (1:500) |
| Antibody | Alexa Fluor 647 Donkey Anti-Chicken IgY (Donkey polyclonal) | Jackson ImmunoResearch, PA, USA | Cat#: 703-605-155 RRID: AB_2340379 | IF (1:500) |

*Appendix 1 Continued on next page*

*Appendix 1 Continued*

| Reagent type (species) or resource | Designation | Source or reference | Identifiers | Additional information |
|---|---|---|---|---|
| Antibody | Alexa Fluor 647 Donkey Anti-Goat IgG (Donkey polyclonal) | Jackson ImmunoResearch, PA, USA | Cat#: 705-605-147 RRID: AB_2340437 | IF (1:500) |
| Antibody | Alexa Fluor 647 Donkey Anti-Guinea pig IgG (Donkey polyclonal) | Jackson ImmunoResearch, PA, USA | Cat#: 706-605-148 RRID: AB_2340476 | IF (1:500) |
| Antibody | Alexa Fluor 647 Donkey Anti-Rabbit IgG (Donkey polyclonal) | Jackson ImmunoResearch, PA, USA | Cat#: 711-605-152 RRID: AB_2492288 | IF (1:500) |
| Antibody | Biotin-SP Donkey Anti-Goat IgG (Donkey polyclonal) | Jackson ImmunoResearch, PA, USA | Cat#: 705-065-147 RRID: AB_2340397 | IF (1:500) |
| Antibody | Biotin-SP Donkey Anti-Guinea pig IgG (Donkey polyclonal) | Jackson ImmunoResearch, PA, USA | Cat#: 706-065-148 RRID: AB_2340451 | IF (1:500) |
| Antibody | Biotin-SP Donkey Anti-Rabbit IgG (Donkey polyclonal) | Jackson ImmunoResearch, PA, USA | Cat#: 711-065-152 RRID: AB_2340593 | IF (1:500) |
| Antibody | Biotin-SP Donkey Anti-Rat IgG (Donkey polyclonal) | Jackson ImmunoResearch, PA, USA | Cat#: 712-065-153 RRID: AB_2315779 | IF (1:500) |
| Antibody | Rhodamine Red-X Donkey Anti-Chicken IgY (Donkey polyclonal) | Jackson ImmunoResearch, PA, USA | Cat#: 703-295-155 RRID: AB_2340371 | IF (1:100) |
| Antibody | Rhodamine Red-X Donkey Anti-Guinea pig IgG (Donkey polyclonal) | Jackson ImmunoResearch, PA, USA | Cat#: 706-295-148 RRID: AB_2340468 | IF (1:100) |
| Antibody | Rhodamine Red-X Donkey Anti-Mouse IgG (Donkey polyclonal) | Jackson ImmunoResearch, PA, USA | Cat#: 715-295-151 RRID: AB_2340832 | IF (1:100) |
| Antibody | Rhodamine Red-X Donkey Anti-Rabbit IgG (Donkey polyclonal) | Jackson ImmunoResearch, PA, USA | Cat#: 711-295-152 RRID: AB_2340613 | IF (1:100) |
| Antibody | Rhodamine Red-X Donkey Anti-Rat IgG (Donkey polyclonal) | Jackson ImmunoResearch, PA, USA | Cat#: 712-295-153 RRID: AB_2340676 | IF (1:100) |
| Sequence-based reagent (*RNAscope probe*) | Mm-Npy-C3 | ACD BioTechne, CA, USA | Cat#: 313321-C3 | |
| Sequence-based reagent (*RNAscope probe*) | Cre-C2 | ACD BioTechne, CA, USA | Cat#: 312281-C2 | |
| Peptide, recombinant protein | [Leu$^{31}$,Pro$^{34}$]-neuropeptide Y | Tocris, Abingdon, UK | Cat#: 1176 | |
| Chemical compound, drug | Clozapine-*N*-oxide (CNO) | Tocris, Abingdon, UK | Cat#: 4936 | |
| Chemical compound, drug | CNO-dihydrochloride | Tocris, Abingdon, UK | Cat#: 6329 | |
| Chemical compound, drug | Gabapentin | Sigma-Aldrich, Glasgow, UK | Cat#: PHR1049-1G | |

*Appendix 1 Continued on next page*

*Appendix 1 Continued*

| Reagent type (species) or resource | Designation | Source or reference | Identifiers | Additional information |
|---|---|---|---|---|
| Chemical compound, drug | SR95531 (Gabazine) | Abcam, Cambridge, UK | Cat#: ab120042 | |
| Chemical compound, drug | Strychnine hydrochloride | Sigma-Aldrich, Glasgow, UK | Cat#: S8753 | |
| Chemical compound, drug | Tetrodotoxin citrate (TTX) | Alamone Labs, Jerusalem, Israel | Cat#: T-550 | |
| Chemical compound, drug | NBQX disodium salt | Abcam, Cambridge, UK | Cat#: ab120046 | |
| Chemical compound, drug | D-APV | Tocris, Abingdon, UK | Cat#: 0106 | |
| Chemical compound, drug | 4-Aminopyridine (4-AP) | Sigma-Aldrich, Glasgow, UK | Cat#: 275875 | |
| Chemical compound, drug | BMS 139885 | Tocris, Abingdon, UK | Cat#: 3242 | |
| Chemical compound, drug | (−)-Bicuculline methobromide | Tocris, Abingdon, UK | Cat#: 0109 | |
| Chemical compound, drug | Chloroquine diphosphate salt | Sigma-Aldrich, Glasgow, UK | Cat#: C6628 | |
| Chemical compound, drug | Complete Freund's adjuvant (CFA) | Sigma-Aldrich, Glasgow, UK | Cat#: F5881 | |
| Software, algorithm | Neurolucida | MBF Bioscience, VT, USA | RRID: SCR_001775 | https://www.mbfbioscience.com/neurolucida |
| Software, algorithm | Neurolucida Explorer | MBF Bioscience, VT, USA | RRID: SCR_017348 | https://www.mbfbioscience.com/neurolucida-explorer |
| Software, algorithm | pClamp | Molecular Devices, CA, USA | RRID: SCR_011323 | https://www.moleculardevices.com/products/axon-patch-clamp-system/acquisition-and-analysis-software/pclamp-software-suite#gref |
| Software, algorithm | Zen Black | Carl Zeiss, Germany | RRID: SCR_018163 | https://www.zeiss.com/microscopy/int/products/microscope-software/zen.html |
| Software, algorithm | Prism | GraphPad Software, CA, USA | RRID: SCR_002798 | https://www.graphpad.com/scientific-software/prism/ |
| Software, algorithm | Behavioral Observation Research Interactive Software (BORIS) | Oliver Friard & Marco Gamba, University of Torino, Italy | RRID: SCR_021434 | https://www.boris.unito.it/ |

