## [Editor Report · eLife assessment]

Boyle et al identify Npy-expressing dorsal horn neurons as powerfully inhibiting pain and itch under normal and pathological conditions. The **valuable** data are **convincing**, and the effect sizes are robust and directly challenge previous work.

---

## [Referee Report · Reviewer #1 (Public Review)]

The first synapses of the pain pathway are concentrated in the superficial spinal cord dorsal horn. Here peripheral inputs are processed by local interneuron circuitry before ascending to the brain. The spinal dorsal horn is organized into lamina with the resident interneurons differentiated by their anatomy, physiological and molecular properties. Over the past decade, the restricted expression of select genes has been used to assign potential function to dorsal horn neuron "cell types". This type of work has relied on the genesis of Cre-reporter mouse strains and intersectional tools to generate mice where select sets of neurons can be activated, inhibited, or ablated. The picture that has emerged from these types of experiments is murky but favors the model where there exist genetically defined cell-types play distinct roles in the detection of painful, itchy, thermal, and mechanical stimuli under normal and pathological situations. The current work by Boyle and colleagues concerns itself with the dorsal horn neurons expressing the neuropeptide NPY. This study is directly related to previously published work that demonstrated that ablating spinal cord neurons that express Npy, including those who express this gene transiently during development, resulted in mice that had heightened touch-evoked itch that seemed different from the canonical chemical itch pathways previously identified. A major conclusion from this previous work was that other modalities were unaffected. Subsequent work built on these findings to identify the potential touch inputs and spinal neuron expressing the Npy receptor as part of a mechanical itch circuit.

This current work by Boyle and colleagues challenge challenges this view by providing evidence that in adult mice, the dorsal horn neurons expressing Npy function to broadly inhibit pain and itch. The authors use direct injection of viral vectors, chemogenetics and synaptic silencing to probe the behavioral effects of stimulating or silencing Npy-expressing dorsal horn neurons in a variety of assays under normal and pathological conditions known to produce allodynia and hyperalgesia. Overall, this is a rather carefully conducted study with the appropriate controls. The data are clear, the effect sizes robust and the presentation easy to follow. These findings challenge the conclusion that these neurons are involved selectively in mechanical itch and instead reveal a potentially clinically important group of neurons for pain.

---

## [Referee Report · Reviewer #2 (Public Review)]

Whether and how molecularly defined neuronal groups in the spinal cord process distinct modalities are of great interest. In this study, Boyle et al. characterized roles of inhibitory neurons expressing NPY in adult mice. By using chemogenetic, electrophysiological tools and behavioral measurements, the authors discovered that activating NPY+ interneurons strongly reduced pruritogen-evoked itch and reflexive behaviors (acute nociception or under inflammation / neuropathic pain states). Silencing NPY+ spinal interneurons enhanced spontaneous and chemical itch in a GRPR+ neurons dependent manner. The authors concluded that, unlike previous findings suggesting that these neurons are selective for mechanical itch, adult NPY+ interneurons play dual roles in gating various types of itch and pain.

The authors performed careful characterization and comparisons between development lineage and adult spinal neurons expressing NPY. This lays the foundation of the current study. The behavioral measurements were also well designed with proper controls.

---

## [Referee Report · Reviewer #3 (Public Review)]

In the present study by Boyle et al., the function of NPY expressing spinal neurons in pain and itch perception is studied. While the function of these neurons has been addressed previously, the difference to previous studies is the combinatorial use of AAV encoded effectors and cre transgenic mice whereas previous studies relied on cre transgenic mice and reporter mice encoding the effector or only viruses. Boyle at al. demonstrate that their strategy enabled them to restrict the analysis to only those neurons expressing NPY in the adult mouse compared to a more heterogenous population that had been studied before. By using a combination of morphology, electrophysiology and behavioral paradigms they convincingly show that NPY neurons impact pruritoception via inhibiting GRPR neurons. Furthermore, they indicate a role of NPY neurons also in nociception as activation attenuates not only responses to acute nociceptive stimuli but also blocks inflammation or nerve injury induced mechanical and heat hypersensitivity. Selectively activating NPY neurons in vivo may therefore be a promising strategy to treat neuropathic pain.

The result of this study extends and partially contrast previous studies. The authors argue that contrasting results may be due to the different experimental strategies (e.g. only neurons expressing NPY adult in the present study versus a more heterogeneous population before).

Overall, the experiments are convincing, and the quality of the data/figures is exceptionally high.

---

## [Author Response]

The following is the authors’ response to the original reviews.

**Reviewer 1 (Recommendations For The Authors):**

The strikingly different conclusion from the previous Bourane study seems to stem from the experimental approaches. Rather than using genetic crosses that target all neurons from the hindbrain and spinal cord that express Npy at any point in development, Boyle et al target their manipulations specifically to the lumbar region of the superficial dorsal horn in adult mice using direct viral injections. Thus, Boyle is almost certainly manipulating much fewer neurons that the original study. How then is their behavioral effects so much greater? At the minimum, the authors need to discuss this discrepancy head on. Better would be a direct molecular/anatomical comparison of the neurons targeted by each approach. This could be done using Nyp-Cre mice crossed to a Rosa-LSL-reporter strain and quantifying the overlap with the same markers used here. Perhaps, the intersectional approach with Lbx1 resulted in labeling of a different population of neurons than the adult AAV injections? Although likely outside the scope, given this work directly questions the main conclusion of the Bourane paper, it will be important to see a replication of the original finding of selectivity to mechanical itch.

We agree that our approach should be manipulating a smaller population of neurons, and that it is therefore suprising that we see greater behavioural effects. Please see our response to "Weakness 1" of Reviewer 2 for consideration of this point. We have already provided a direct molecular comparison as requested by the reviewer, and this appears in Figure 1 supplement 1. Here we used tissue from NPY::Cre that had been crossed with Ai9 mice (i.e. a Rosa-LSL-reporter) and had received intraspinal injections of AAV.flex.GFP. We then characterised the neurochemistry of tdTomato+ cells that were GFP+ or GFP-negative.

2. The authors state that, "91.6% ± 0.3% of cells classed as Cre-positive cells were also Npy-positive, and these accounted for 62.1% ± 0.6% of Npy-positive cells" If I am reading this correctly, does that mean that 40% of the Npy+ cells are Cre negative? If so, how is this possible?

This interpretation is correct. For quantification of RNAscope data we used a cut-off level of 4 transcripts, and cells with fewer than 4 transcripts were classed as negative. It is likely that some of the NPY cells classified as negative for Cre would have had some Cre mRNA (sufficient to cause recombination), but at a level below this threshold. It is also possible that some NPY+ cells would fail to express Cre, since this is a BAC transgenic mouse, rather than a knock-in.

3. Similarly, the authors state that "great majority of FP-expressing neurons in laminae I-III were immunoreactive (IR) for NPY (78.5% ± 3.6%), and these accounted for 74.6% ± 109 1.9% of the NPY-IR neurons in this area". So does this mean 20% of the recombination is non-specific/in other cell types that could be involved in pain/itch sensation?

Our finding that 91.6% of cells with Cre mRNA were also positive for Npy mRNA (see above) indicates that Cre expression was largely restricted to NPY cells. The failure to detect NPY peptide in some of these cells probably results from the relatively low level of peptide seen in the cell bodies of peptidergic neurons, which results from the rapid transport of peptides into their axons.

4. Comparing Fig 3B and Fig4B it seems the control baseline von Frey responses are different. In fact, baseline response in Fig4b is quite like the CNO effect in Fig 3B. Unless I'm misunderstanding something, this seems quite odd?

We agree that there is a difference between the baseline responses. We are not aware of any particular reason for this, and we think that it reflects a degree of variability that is seen with the von Frey test. Interestingly, the baseline values for the SNI cohort (Fig 4E) lies between the values in Fig 3B and Fig 4B.

5. In Fig 4E, the behavior of the CNO treated mice is quite variable. Can the authors comment as to how this might be happening? Does the effect correlate with viral transduction?

We did not see any obvious correlation between the extent of viral transduction and the behaviour of individual mice.

6. Fig6, the PDyn-Cre experiment, is a bit of a non sequitur?

Please see our response to "Weakness 2" of Reviewer 2 for consideration of this point.

7. The conclusion is unusually long. I recommend trimming it to make it more concise.

We presume that this refers to the Discussion. However, this was ~1550 words, and we do not feel that that is unusually long.

**Reviewer 2 (Public Review):**

Weaknesses

There is inadequate discussion about previous studies of NPY interneurons. Specifically, the authors should address why a more restricted subset of these neurons (this study) have broader effects than seen previously.

We have expanded the discussion on the discrepancies between our findings and those reported previously. We state at the outset that we are targeting a more restricted population (lines 509-10), and we now go into more detail concerning both similarities and differences between our findings and the reasons that we think may underlie any discrepancies (various changes between lines 522-575).

2. I cannot see the reason for including results from manipulation of Dyn+ interneurons in this paper. First, the title does not reflect roles of spinal Dyn+ population. In addition, without further experiments characterizing relationships between NPY and Dyn interneurons in modulating itch and/or nociception, Dyn datasets seem to deviate from the main theme.

We had previously shown that activating Dyn-INs suppressed pruritogen-evoked itch (Huang et al 2018), but it was important to test whether silencing these cells would have the opposite effect. Our finding of overlap in function (i.e. both NPY-INs and Dyn-INs suppress itch, and that both innervate GRPR cells) provides strong evidence against the idea that neurochemically-defined interneuron populations have highly specific functions, and we now state this in the Discussion. The anatomical experiments (which follow on from the functional studies) provide important new information concerning synaptic circuitry of the dorsal horn, by showing that NPY-INs preferentially innervate GRPR cells, and provide around twice as many synapses on these cells, compared to the Dyn-INs. Interestingly, this correlates with the relatively large optogenetically-evoked IPSCs that we saw when NPY-INs were activated, compared to those reported by Liu et al (2019) when galanin-expressing (which largely correspond to Dyn-INs) were activated. By including these findings in the paper, we are able to make comparisons between these two populations.

3. While the authors provided convincing evidence that GRPR+ neurons serve as a downstream effector of NPY+ neuron evoked itch, the relationship between GRPR and NPY neurons in modulating pain is not examined. Therefore, Fig. 7B is pure speculation and should be removed.

We feel that our recent findings that GRPR neurons correspond to vertical cells, that they respond to noxious stimuli, and that activating them results in pain-related behaviours, makes it reasonable to speculate that the NPY/GRPR circuit may also be involved in the anti-nociceptive action of NPY cells. The legend for Fig 7B already refers to this as a "potential circuit", and we have toned down the corresponding part of the discussion to say that our findings "raise the possibility" that this is the case (lines 605-7). We feel that this part of the figure is important, as otherwise our summary diagram ignores some of the main findings of the paper, and we hope that this is now acceptable.

Recommendations For The Authors

Fig. 1G: the "misexpression" of tdTomato neurons was much more prominent in deep dorsal horn laminae but not in the superficial ones. Was this representative? Can the authors perform a laminae specific characterization?

We did test for this possibility in 2 NPY::Cre;Ai9 mice that had received intraspinal injections of AAV.flex.GFP, and found that there was a modest difference - 62% of tdTomato+ cells in laminae I-II, but only 39% of those in lamina III, were GFP+. This suggests that "misexpression" may have differed slightly between these regions. However, since the difference was quite modest, and we were only able to analyse tissue from two mice in this way, we did not include these findings in the paper.

2. I have a lot of problems interpreting the c-Fos data in Fig. 2 E and F. For the mCherry- population, how was the quantification performed? From the image, it does not look like 2030% of cells express c-Fos; at a minimum a clear stain of neurons would be needed. Similarly, the identification of NPY cells is not particularly convincing (e.g., middle arrowhead lower 2 panels in C).

We have provided further details on how the analysis was performed (changes made to lines 1016-29). NeuN staining was used to reveal all neurons, and a modified optical disector method was performed from somatotopically appropriate regions of the dorsal horn. As noted by the Reviewer, NeuN staining was required to allow identification of mCherrynegative cells. However, we have not included the NeuN immunoreactivity in the image, as this would add considerably to the complexity. These images are from single optical sections, and therefore the overall numbers of cells are low (in comparison to what would be seen in a projected image). The intensity of mCherry staining varied between cells. However, for all mCherry-positive cells (including the example referred to by the Reviewer), there was clear staining in the membrane, which could be followed in serial sections.

3. Please add individual data points for all quantifications.

These have been added.

**Reviewer 3 Recommendations For The Authors:**

It is somewhat surprising that there is no effect on CPP after activating spinal NPY neurons in neuropathic mice, given the almost complete rescue of hypersensitivity to baseline values in the nociceptive tests. Based on the methods, it appears that conditioning was carried out already 5 min after CNO injection. Yet, suppression of c-fos activity in excitatory spinal dh neurons was observed 30min after CNO injection. Also, it is not clear to me when CNO was injected prior to the nociceptive or CQ testing?

Have the authors considered that conditioning from 5-35 min after CNO injection might be too short after CNO injection to achieve a profound analgetic effect?

In a previous study (Polgár et al 2023), we had observed the timecourse of CNO-evoked itch and pain behaviours in mice in which GRPR cells expressed hM3Dq. We found that these started within 5 minutes of i.p. CNO injection (e.g. Fig S2 in that paper). In addition, the timecourse of action of gabapentin and CNO (both given i.p.) are likely to be similar, and there was a preference for the chamber paired with gabapentin. We are therefore confident that the conditioning period with CNO was adequate. We now explain this in the Methods section (lines 846-52). The timing of CNO injections for the nociceptive and CQ tests is now described (lines 749-55).

2. The authors claim that tonic pain was not affected based on the conditioned place preference test. Efficacy in withdrawal response tests and in the CPP differ by more than duration of the stimulus. I'd suggest using more cautious wording here.

We agree that caution is needed in interpreting the results of the CPP experiments. We have therefore replaced "does" with "may" in the Results section (line 336) and "did" with "may" in the Discussion (line 620).

3. On page 9 the authors state "...suggesting that they suppress the transmission of pain- and itch-related information in the dorsal horn." However, pain is not affected in the loss of function experiments suggesting some qualitative differences in the role of the NPY neurons in itch and pain. This should also be reflected more clearly in this statement and in the discussion e.g. "suppress itch" and "can suppress pain".

We accept the point made by the Reviewer. We have slightly altered the wording in lines 249-51 and 610 to reflect this.